# The Atlas of In-Context Learning: How Attention Heads Shape In-Context Retrieval Augmentation

**Patrick Kahardipraja**[1,*]    **Reduan Achtibat**[1,*]    **Thomas Wiegand**[1,2,3]

**Wojciech Samek**[1,2,3,†]    **Sebastian Lapuschkin**[1,4,†]

[1]Department of Artificial Intelligence, Fraunhofer Heinrich Hertz Institute
[2]Department of Electrical Engineering and Computer Science, Technische Universität Berlin
[3]BIFOLD - Berlin Institute for the Foundations of Learning and Data
[4]Centre of eXplainable Artificial Intelligence, Technological University Dublin
{firstname.lastname}@hhi.fraunhofer.de

## Abstract

Large language models are able to exploit in-context learning to access external knowledge beyond their training data through retrieval-augmentation. While promising, its inner workings remain unclear. In this work, we shed light on the mechanism of in-context retrieval augmentation for question answering by viewing a prompt as a composition of informational components. We propose an attribution-based method to identify specialized attention heads, revealing in-context heads that comprehend instructions and retrieve relevant contextual information, and parametric heads that store entities' relational knowledge. To better understand their roles, we extract function vectors and modify their attention weights to show how they can influence the answer generation process. Finally, we leverage the gained insights to trace the sources of knowledge used during inference, paving the way towards more safe and transparent language models.

## 1 Introduction

Many, if not most language tasks can be framed as a sequence to sequence problem [61, 69]. This view is integral to how modern Large Language Models (LLMs) operate, as they are able to approximate relations between an input and an output sequence not only as a continuation of text, but also as a response to a stimulus [64]. In a sense, input prompts serve as a query to search and induce function(s) in a vast, high-dimensional latent space, where the corresponding process can be cast as question answering [50] or instruction following [56, 80].

This capability is brought forth with the introduction of in-context learning (ICL) [12] that enables LLMs to adapt to new tasks with few demonstrations at inference time, without additional fine-tuning. Previous work has investigated ICL from various perspectives, including its relation to induction heads that can replicate token patterns during prediction [54], the ability to compress attention heads to function vectors (FVs) representing a specific task [31, 74], and how it can emerge when transformers [75] implicitly learn to perform gradient-based optimization [4, 77]. Besides meta-learning, ICL can be used for *retrieval-augmentation* [62], where external knowledge from web retrieval corpora [11, 29, 33] or dialogue information [66, 88, 92] is given instead of input-output pairs to ground LLMs during generation. However, the mechanism behind ICL for knowledge retrieval is not yet fully understood. In this work, we aim to shed light on this question.

---

[*]Equal contribution.
[†]Corresponding authors.

39th Conference on Neural Information Processing Systems (NeurIPS 2025).

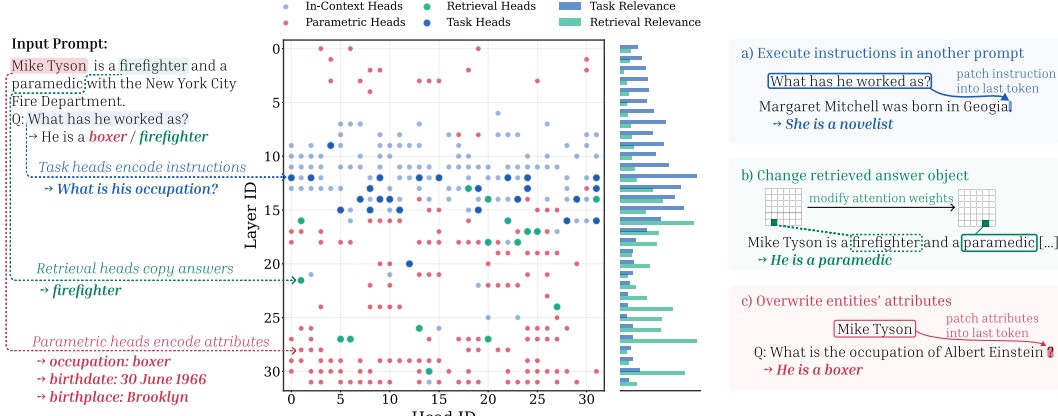

Figure 1: Functional map of in-context and parametric heads in Llama-3.1-8B-Instruct. They are surprisingly well-structured and operate on the input prompt at various levels, with in-context heads processing information in the prompt, including instruction comprehension and retrieval operations — and parametric heads that encode relational knowledge. In-context heads can specialize to task heads to parse instructions (blue) or retrieval heads for verbatim copying (green). Together with parametric heads, they affect the answer generation process through function vectors that they transport (**a**, **c**) or their attention weights (**b**). Our relevance analysis (bar plot) shows that instruction-following capabilities emerge in middle layers, while answer retrieval occurs in later layers. Details in C.1.

On question answering tasks, we show that by viewing a prompt as a composition of informational components, certain attention heads perform various operations on the prompt at different stages of inference and layers (Figure 1). Our method identifies two groups of heads based on their functions: *parametric heads* that encode relational knowledge [27, 57] and *in-context heads* responsible for processing information in the prompt. Further, as in-context heads need to understand which prompt components to process and how, we hypothesize that they specialize to fill their respective roles.

Our analysis shows that in-context heads can indeed execute specialized functions such as instruction comprehension and retrieval of relevant contextual information. To investigate this further, we curate a controlled biography dataset with entities extracted from Wikidata [78]. Remarkably, we find that through compressing them to FVs or modifying their weights, both in-context and parametric heads can induce specific, targeted functions.

Building on these insights, we probe for sources of knowledge used during retrieval-augmented question answering and show where it is localized within the input context. Our attempt shows promising results, serving as an initial step towards more transparent retrieval-augmented generation (RAG) systems. Overall, our contributions can be summarized as follows:

- We describe an attribution-based method to determine attention heads that play a key role during in-context retrieval augmentation for question answering, revealing that they operate on the prompt in a distinct manner. Our method can thus be used to reliably categorize attention heads into in-context and parametric heads.

- We analyze how in-context heads specialize in reading instructions and retrieving information, mapping their location across model layers. Additionally, we demonstrate the influence of in-context and parametric heads on the answer generation process, by compacting them into function vectors or modifying their attention weights.

- We present preliminary results on enabling causal tracing of source information for retrieval-augmented LMs, suggesting fruitful directions for interpretability of RAG systems.

## 2 Related Work

**Retrieval Augmentation** Retrieval-augmented language models (RALMs) [40, 45] address inherent limitations of LMs by providing external knowledge sources. Despite this advantage, issues such as discrepancies between contextual and parametric knowledge may occur [48, 87]. Some

works have studied mechanisms behind knowledge preference in RALMs [51, 55, 91], but they focus on simple domains. On the other hand, those that explored more complex domains mostly only analyzed RALMs' behavior at the output level [14, 71, 85]. Closely related to our work is that of Jin et al. [37], where the authors also discovered in-context and parametric heads. Compared to them, we show that interactions among in-context heads are subtly more complex, since they can specialize into task or retrieval heads. Our approach also allows to demonstrate their roles in shaping the model's representation along with parametric heads, by transforming them into FVs or modifying their respective weights.

Another disadvantage of RALMs is that they cannot guarantee faithful answer attribution[3] to contextual passages [25], which necessitates a shift to interpretability. Recent efforts to address this include leveraging contrastive gradient attribution [60, 89], fitting surrogate models [16], and treating attention weights as features [17]. In relation to this, we train a probe based on retrieval heads to track for knowledge provenance.

**The Role of Attention**  Attention mechanisms have been previously observed to encode many kinds of information. Clark et al. [15] showed that they correspond well to linguistic properties such as syntax and coreference. Similarly, Voita et al. [76] found that syntactic heads play an important role in machine translation models. In relation to world knowledge, Geva et al. [27] proposed that certain heads store factual associations and demonstrated how they extract an attribute of a given subject-relation pair. Interestingly, attention also appears to encode a sense of "truthful" directions [46]. With the exception of Voita et al. [76], the above works make use of attention weights, which might not fully capture the model's prediction [10, 35, 81]. Our work can be seen as an attempt to reconcile both perspectives: analyses based on attention weights and feature attribution methods [9].

**In-Context Learning**  Numerous works have studied ICL since its introduction. Liu et al. [47] studied what constitutes a good example for demonstrations. Dai et al. [20] suggested that ICL can be understood as an implicit fine-tuning. ICL is a general phenomenon, although it is commonly assumed to be unique to autoregressive models [63]. At the component level, ICL is primarily associated with induction heads [21, 54]. However, recent findings showed that certain heads can also be compressed to form FVs that represent a demonstrated task [31, 74]. Yin and Steinhardt [90] investigated the connection between these heads and induction heads, showing that they are distinct from each other and how models' ICL performance is mainly driven by the former. ICL can also be viewed as a mixture of meta-learning and retrieval [52]. In that regard, we study the latter perspective to understand its mechanism as a specific instantiation of ICL, with a focus on the retrieval augmentation paradigm.

## 3   Background and Preliminaries

The self-attention mechanism in transformers poses a challenge in understanding which heads actually contribute during in-context retrieval augmentation, and how they process various components in the prompt. This is mainly due to the fact that information from different tokens gets increasingly mixed as layers go deeper and how several attention heads may implement redundant functions [79]. A natural option is to analyze attention weights, as they are an inherent part of a model's computation. However, attention can be unfaithful [34], which questions its validity as an explanation [10, 65]. This problem is further exacerbated by "attention sinks" [43, 84] — a phenomenon where heads heavily attend to the first token and obscure the weights of other tokens in the sequence.

An alternative would be to use feature attribution methods [9], as they trace the contribution of each input element to the model's output prediction. Propagation-based feature attribution [8, 67, 68] especially takes the entire computation path into account, which can be used to characterize attention heads [76] or identify latent concepts [1]. Furthermore, feature attribution is able to estimate causal relations [26] *e.g.,* to automate circuit discovery [18, 22, 30, 32, 70], and thus enables to observe how a specific attention head affects a model's prediction.

In this section, we provide a description of AttnLRP [2], on which our method is based, due to its superior performance and efficiency in transformer architectures compared to other attribution methods. We also provide an overview of the multi-head attention mechanism in transformers, which

---

[3]Here, the term *answer attribution* means the use of external documents to support the generated response, which is different from *feature attribution* used throughout this work to describe interpretability techniques.

we leverage through AttnLRP to identify both in-context and parametric heads (§5). Additionally, we analyze the specialization of in-context heads, show causal roles of the identified heads (§6), and use this information for reliable and efficient source tracking of facts in retrieval-augmented LMs (§7).

## 3.1 Layer-wise Relevance Propagation

Feature attribution methods aim to quantify the contribution of input features $\mathbf{x}$ to the overall activation of an output $y$ in linear but also highly non-linear systems. We define a function $\mathcal{R}$ that maps the input activations $\mathbf{x}$ to relevance scores indicating their causal effect on a model's output logit $y$:

$$\mathcal{R} : \mathbb{R}^N \times \mathbb{R} \rightarrow \mathbb{R}^N, \qquad (\mathbf{x}, y) \mapsto \mathcal{R}(\mathbf{x} \mid y).$$

In principle, any feature attribution method can be employed for $\mathcal{R}$, though trade-offs between faithfulness and computational efficiency must be carefully considered. Perturbation-based approaches [49] typically offer high faithfulness but incur exponential computational costs, as each ablated latent component requires at least one dedicated forward pass [30]. In contrast, gradient-based methods [67] are computationally more efficient, requiring only a single backward pass, which makes them well suited for large-scale interpretability studies. However, they are susceptible to noisy gradients, which are distorted by non-linear components such as layer normalization [5, 86]. To address these limitations, we adopt AttnLRP [2], an extension of Layer-wise Relevance Propagation (LRP) [8] designed for transformer architectures. As a backpropagation-based technique, AttnLRP propagates relevance scores from a model output to its inputs in a layer-wise manner and can be implemented efficiently via modified gradient computation in a single backward pass. Importantly, it incorporates stabilization procedures for non-linear operations, thereby improving the faithfulness of relevance distributions compared to standard gradient- or other decomposition-based methods [7].

Relevance scores produced by (Attn)LRP can be either positive or negative. Positive relevance indicates an amplifying effect on the final model logit $y$, whereas negative relevance reflects an inhibitory influence. Without loss of generality, we focus our analysis on signal-amplifying components by considering only positive relevance scores. Formally, we define:

$$\mathcal{R}^+(\mathbf{x}|y) = \max \left( \mathcal{R}(\mathbf{x}|y), 0 \right) \tag{1}$$

This yields a clearer separation between in-context and parametric heads in the subsequent analysis.

## 3.2 Attention Mechanism

While the original formulation of the multi-head attention mechanism [75] concisely summarizes the parallel computation of attention heads, our goal is to isolate their individual contributions. To this end, we reformulate the equations to make the influence of each head more explicit [21, 23]. Let $\mathbf{X} = (\mathbf{x}_1, \ldots, \mathbf{x}_S) \in \mathbb{R}^{d \times S}$ denote the matrix of hidden token representations for a sequence of length $S$ with dimension $d$, and suppose our model employs $H$ parallel heads, each of dimension $d_h = d/H$. Then, the computation of the multi-head attention layer can be reformulated into $H$ complementary operations, where each head $h$ produces an intermediate attention output $\mathbf{z}_i^h \in \mathbb{R}^{d_h}$:

$$\mathbf{z}_i^h = \sum_{j=1}^{S} \mathbf{A}_{i,j}^h \left( \mathbf{W}_V^h \, \mathbf{x}_j \right) \tag{2}$$

where $\mathbf{A}_{i,j}^h$ is the attention weight of token $i$ attending to token $j$, and $\mathbf{W}_V^h \in \mathbb{R}^{d_h \times d}$ is the per-head value projection. The final output is obtained by multiplying the intermediate output of each head with their corresponding output projection matrix $\mathbf{W}_O^h \in \mathbb{R}^{d \times d_h}$, followed by summing:

$$\hat{\mathbf{x}}_i = \sum_{h=1}^{H} \mathbf{W}_O^h \, \mathbf{z}_i^h \tag{3}$$

We leverage the multi-head attention mechanism in transformers through the lens of AttnLRP to identify both in-context and parametric heads in §5 and how in-context heads specialize in §6.

## 4 Experimental Setup

**Models** We use instruction-tuned LLMs due to their increased capability on question answering (QA) tasks in our preliminary experiments: Llama-3.1-8B-Instruct [28], Mistral-7B-Instruct-v0.3 [36],

and Gemma-2-9B-it [72]. We apply AttnLRP based on their `huggingface` implementations [82]. For the rest of this work, we refer to each model by their family prefix.

**Datasets**  To perform our analyses, we use two popular open-domain QA datasets: NQ-Swap [48] and TriviaQA (TQA) [38]. NQ-Swap is derived from Natural Questions [44], a QA dataset with questions collected from real Google search queries and answer spans annotated in Wikipedia articles. TQA contains trivia questions with answers sourced from the web. Both datasets are based on the MRQA 2019 shared task version [24].

Similar to Petroni et al. [58], we consider different types of contextual information to see how they affect in-context and parametric heads. We use oracle contexts as they are always relevant to the question and contain the true answer. In addition, we use counterfactual contexts as they contain information that is usually not seen during pre-training and fine-tuning stages, thus forcing models to rely on the text to answer the question correctly. Oracle context is often not available; therefore we also use Dense Passage Retriever (DPR) [39] with a Wikipedia dump from December 2018 as our retrieval corpus. For simplicity, we only select the top one retrieved document. We show results for oracle and counterfactual contexts in the main paper and retrieved DPR contexts in Appendix B.

Inspired by Allen-Zhu and Li [6], we build a human biography datasets to allow us to better understand the characteristic of in-context and parametric heads and conduct controlled experiments. Using Wikidata [78], we collect profiles for random 4,255 notable individuals containing their date of birth, birth place, educational institute attended, and occupation. We concatenate the attributes of each individual in a random order to form a biographical entry and ask Llama-3.1-8B-Instruct to paraphrase it. See Appendix A for more details.[4]

## 5   Localization of In-Context and Parametric Heads

In retrieval-augmented generation, LLMs are faced with the option to generate responses by using a wealth of knowledge they learn during pre-training or by relying on contextual information provided in the prompt through ICL. Here, we categorize attention heads that are responsible for both capabilities.

**Method**  We aim to identify the sets of in-context heads $\mathcal{H}_{\text{ctx}}$ and parametric heads $\mathcal{H}_{\text{param}}$ as depicted in Figure 1. We define in-context heads as those that mainly contribute to the model's prediction during RAG by using contextual information, whereas parametric heads primarily contribute upon reliance on internal knowledge. We hypothesize that each head type contributes maximally under a specific condition while having minimal influence in others, *i.e.,* in-context heads are expected to contribute the most in open-book settings and the least in closed-book settings, and vice versa. We analyze questions with *counterfactual* contexts, forcing retrieval to produce a counterfactual prediction $y_{\text{cf}}$ that disagrees with the parametric answer. Conversely, we also focus on closed-book settings where contextual information is minimized, to identify parametric heads and reduce the chance that in-context heads contribute. We restrict our analysis to instances where a gold reference answer $y_{\text{gold}}$ is predicted, to ensure that relevance attribution reflects genuine parametric behavior.

We use AttnLRP to quantify the contribution of each attention head $h$ to the prediction by summing the positive relevance scores assigned to its latent output $\mathbf{z}^h$ across its dimension $d_h$ and over all $i$ token positions, when explaining the targeted prediction $y_t$, which can be either a gold reference answer $y_{\text{gold}}$ or a counterfactual output $y_{\text{cf}}$, depending on the setting:

$$\mathcal{R}^h(y_t) = \sum_{i=1}^{S} \sum_{k=1}^{d_h} \mathcal{R}^+(\mathbf{z}_i^h \mid y_t)_k \in \mathbb{R}. \tag{4}$$

To contrast heads across settings, we compute a difference score $\mathcal{D}$ representing their average contribution in open-book versus closed-book conditions for all $N_h$ heads in the model:

$$\mathcal{D} = \left\{ \mathbb{E}_{X_{\text{OB}}}\left[\mathcal{R}^h(y_{\text{cf}})\right] - \mathbb{E}_{X_{\text{CB}}}\left[\mathcal{R}^h(y_{\text{gold}})\right] \ : \ h = 1, 2, \ldots, N_h \right\} \tag{5}$$

We then identify the most distinctive heads for each behavior by selecting the top 100 heads (around 10%-15% of total heads) with the highest positive and lowest negative difference scores:

$$\mathcal{H}_{\text{ctx}} = \{\text{argsort}_{\text{desc}}(\mathcal{D})\}_{n=1}^{100}, \qquad \mathcal{H}_{\text{param}} = \{\text{argsort}_{\text{asc}}(\mathcal{D})\}_{n=1}^{100} \tag{6}$$

---

[4]Our implementation is publicly available at `https://github.com/pkhdipraja/in-context-atlas`

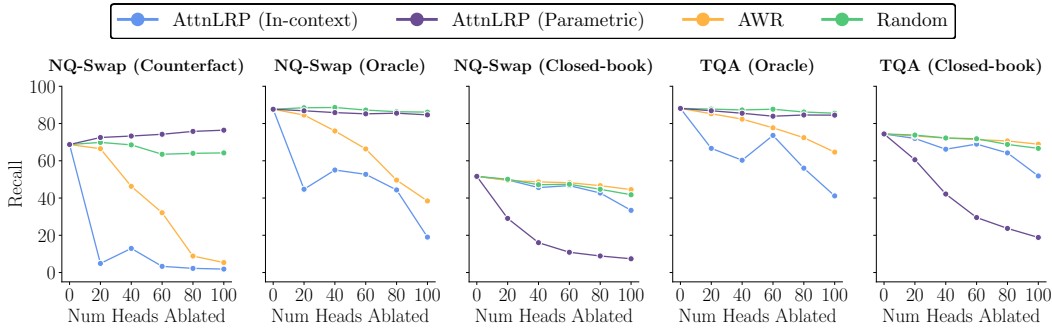

Figure 2: Recall analysis for Llama 3.1 when either in-context or parametric heads are ablated. Removing identified in-context heads noticeably affects the model's performance in open-book QA across various configurations. Conversely, removal of identified parametric heads most strongly affects the model's closed-book QA capabilities. Compared to Wu et al. [83] that only yield AWR (retrieval) heads, our method allows to obtain both in-context and parametric heads.

**Experiments**   To ensure that the identified in-context and parametric heads play a role in QA tasks, we ablate both sets of heads and measure performance drops in settings where they are expected to be mostly active (open- and closed-book, respectively). We also measure if the removal of in-context heads affects the models' capabilities to answer in closed-book setting and vice versa, since this informs to what extent both sets of heads are *dependent* on each other. Furthermore, we want to know if the identified in-context and parametric heads can *generalize* to other datasets. To test this, we compute the score of both heads only over NQ-Swap and reuse the same sets of heads on TQA. To evaluate the aforementioned criteria, we report recall [3] as instruction-tuned models tend to produce verbose responses. As baselines, we select random heads, and also adopt the Attention Weight Recall (AWR) method based on attention maps' activations, as described in [83].

**Results**   We show results for Llama 3.1 here and other models in Appendix B. Figure 2 shows how the recall score evolves when either heads identified as in-context or parametric heads are ablated. We observe that the removal of 20 heads (100 heads) reduces the performance by 13.86%-63.84% (44.26%-68.66%) for open- and closed-book settings across different configurations, indicating the causal influence these heads have on the answers' correctness. Moreover, the performance drops on TQA hold even though the heads are computed on NQ-Swap, showing that the identified in-context and parametric heads are transferrable to other datasets.

We compare in-context heads as identified with our method against AWR heads, and find that the removal of 20 in-context heads results in a roughly similar reduction of recall as removing 100 AWR heads. Furthermore, ablating in-context heads yield a more drastic performance decrease compared to the removal of AWR heads, suggesting that our method is more suitable than those based on attention scores alone to study heads that contribute to response generation. Ablating randomly chosen heads barely affects the model's ability to answer correctly.

We examine whether in-context and parametric heads are independent of each other. As expected, ablating parametric heads has little influence to the model's performance in our open-book setting. Interestingly, this leads to a slight performance increase on NQ-Swap with counterfactual contexts, which suggest that the ablation forces the model to rely more on the given context instead of its own parametric knowledge. Surprisingly, ablating in-context heads in the closed-book setting incurs a non-negligible performance reduction. This is likely due to the influence in-context heads have when processing the input prompt. We explore this in §6.

### 5.1   Resolving Knowledge Conflicts

We further test whether targeted ablation of parametric and in-context heads also helps to resolve conflicts between contextual and parametric knowledge sources. On NQ-Swap with counterfactual contexts, we selectively ablate heads from each category. Removing in-context heads forces the model to rely more heavily on parametric memory, leading to substantial gains in recall with respect to the original, non-counterfactual answers. Conversely, ablating parametric heads promotes stronger

grounding in the provided context as shown before, improving recall for counterfactual open-book answers. We present the full results in Appendix E.

# 6 Functional Roles of In-Context and Parametric Heads

Given that ablating in-context heads yields a non-negligible drop in closed-book QA performance, where no external documents are available, we posit that in-context heads not only process the context but also interpret the *intensional frame* – the semantic structure imposed by the instruction itself [64]. In the counterfactual example below, the intensional frame (the question prompt) is shown in *italics*, the object instance in **bold**, and two equally plausible answers in color:

> "[Mike Tyson was a **firefighter** from 1980 to 1984 with the New York City Fire Department. . . ] Q: *What has Mike Tyson worked as?* A: boxer / firefighter"

To answer correctly, the model must map the intensional frame onto the knowledge triple $(s, r, o^*)$, where $s$ is the *subject* ("Mike Tyson"), $r$ is the *semantic relation* (the predicate specified by the question, here "has worked as"), and $o^*$ is the *object* (the yet to be determined answer, "boxer" or "firefighter"). Depending on where the answer resides, $o^*$ may be retrieved from the model's parametric memory ($o^p$) or from the context ($o^c$). By treating $(s, r, o^*)$ as the complete task specification, we analyze how in-context and parametric heads specialize both to comprehend the intensional frame and to retrieve the object $o^*$ needed to generate the correct answer.

## 6.1 Disentangling the Functionality of In-Context Heads

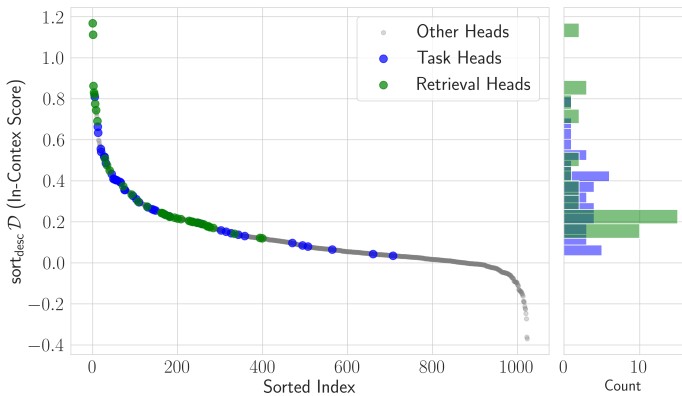

Figure 3: Sorted in-context scores for 1024 heads of Llama 3.1, comparing open-book and closed-book settings via score $\mathcal{D}$. Positive scores indicate in-context behavior, while negative scores reflect parametric behavior. Retrieval heads (green) and task heads (blue) are predominantly high-scoring in-context heads. See Appendix Figure 7 for other models.

Our goal is to identify heads specialized in processing the *intensional frame* and those specialized in retrieving the *answer object* from the context. Inspired by the work of [1], which demonstrates that relevance is effective for separating functional components in latent space, we measure how much relevance of an attention head is assigned to the question and retrieved answer tokens.

**Method** For each head $h$, we compute the total relevance attributed to the attention weight $A_{i,j}^h$ when explaining the logit output $y_t$. Since relevance flows backwards from the output to the input, our goal is to obtain relevance at the input level of each layer. Given that each head transfers information from the *key* at position $j$ to the *query* at position $i$, we aggregate this backward relevance over all possible query positions $i$ to obtain a single relevance score for the source token at key position $j$:

$$\rho_j^h = \sum_{i=1}^{S} \mathcal{R}^+ \left( A_{i,j}^h \mid y_t \right) \tag{7}$$

Here, $\rho_j^h$ represents the total relevance assigned at head $h$ to token $j$ when contributing to logit $y_t$. Next, we aggregate the relevance scores separately for two sets of token positions within the context:

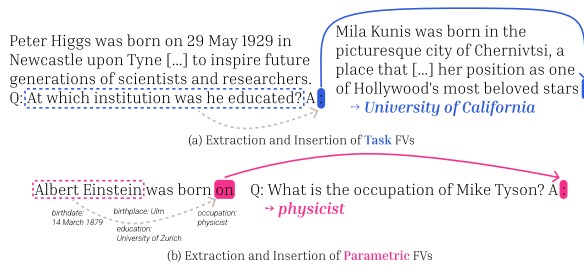

(a) Extraction and Insertion of Task FVs

(b) Extraction and Insertion of Parametric FVs

Figure 4: Extraction and insertion of task and parametric FVs. The induced generation is highlighted in italic.

Table 1: Zero-shot recall scores for task, parametric, and retrieval heads.

| Models | $\mathcal{H}_{\text{task}}^{40}$ | $\mathcal{H}_{\text{param}}^{50}$ | $\mathcal{H}_{\text{ret}}^{40}$ |
|---|---|---|---|
| Llama 3.1 (random) | 18.00 | 6.68 | 15.94 |
| + FVs / Attn Weight | **94.75** | **38.84** | **93.45** |
| Mistral v0.3 (random) | 9.50 | 12.95 | 8.56 |
| + FVs / Attn Weight | **88.50** | **44.04** | **97.03** |
| Gemma 2 (random) | 7.50 | 6.79 | 3.89 |
| + FVs / Attn Weight | **88.00** | **34.77** | **87.36** |

the intensional-frame tokens, denoted as $j \in J_{\text{task}}$, which comprise the question token positions, and the answer object tokens, denoted as $j \in J_{\text{ret}}$, which represent positions of the retrieved object.

$$\rho_{\text{task}}^h = \sum_{j \in J_{\text{task}}} \rho_j^h, \quad \rho_{\text{ret}}^h = \sum_{j \in J_{\text{ret}}} \rho_j^h. \tag{8}$$

Finally, to obtain the sets of specialized task and retrieval heads, we rank heads by their aggregated relevance and select the top $K$, a hyperparameter determined separately for each experiment.

$$\mathcal{H}_{\text{task}}^K = \left\{ \text{argsort}(\rho_{\text{task}}^h)_{\text{desc}} \right\}_{n=1}^K, \quad \mathcal{H}_{\text{ret}}^K = \left\{ \text{argsort}(\rho_{\text{ret}}^h)_{\text{desc}} \right\}_{n=1}^K. \tag{9}$$

**Results**  We compute the task relevance score $\rho_{\text{task}}^h$ and the retrieval relevance score $\rho_{\text{ret}}^h$ over NQ-Swap with counterfactual contexts to minimize influences of parametric heads, and aggregate their distributions across the model layers. In Figure 1, we observe that $\rho_{\text{task}}^h$ initially increases in the early layers where few parametric heads are located, suggesting that early parametric heads enrich the question with relational knowledge. The relevance peaks in the middle layers, where in-context heads dominate, aligning with the transition to a more context-dependent reasoning. In contrast, the retrieval relevance score $\rho_{\text{ret}}^h$ peaks in deeper layers, reflecting the point where the model extracts the final answer object $o^c$. Figure 3 further illustrates the sorted average difference $\mathcal{D}$ between open-book and closed-book settings for all heads, alongside the top 40 task heads $\mathcal{H}_{\text{task}}$ and retrieval heads $\mathcal{H}_{\text{ret}}$. We observe that the highest-scoring in-context heads are primarily composed of retrieval and task heads, emphasizing their role for retrieval augmented generation.

## 6.2 Causal Effects of In-Context and Parametric Heads

An important question is whether the heads we identify truly reflect their assigned functionalities. We examine this under controlled conditions and investigate their causal effects on the answer generation.

**Experiments**  We conjecture that task heads $\mathcal{H}_{\text{task}}$ encode the intensional frame $(s, r, o^*)$ and that parametric heads $\mathcal{H}_{\text{param}}$ contain information of the subject $s$, which depending on the training data may or may not include $o^p$. On the other hand, retrieval heads $\mathcal{H}_{\text{ret}}$ search for $o^c$, allowing them to copy any tokens from the context verbatim, without being restricted to only plausible answers. For task and parametric heads, we compute FVs[5] for each head and insert them into various settings to trigger the execution of their functions. Following Wu et al. [83], we opt for a needle-in-a-haystack (NIAH) setting for retrieval heads and determine their ability to retrieve relevant information from the context by modifying the attention weights. To isolate these behaviors, we conduct our analysis on the biography dataset (§4) and measure recall [3]. For comparison, we also consider random heads for FV extraction and attention modification. See Appendix C for additional results and details, including the selection of hyperparameter $K$.

**Task Heads**  We demonstrate that task heads encode intensional frames. In a zero-shot manner, we extract task FVs from each head in $\mathcal{H}_{\text{task}}$ for four questions relating to all recorded attributes from the biography dataset. Then, we insert them to another biographical entry without a question at the final token position, and also for all subsequent token generations (Figure 4, top). We examine whether

---

[5]A FV can be defined as a sum of averaged heads outputs over a task [74] or computed individually [31]. Following the latter, we consider a FV to be an output of a task or parametric head scaled by scalar $\alpha$.

they reliably induce responses aligned with the original question. In Table 1 (left), we show that applying FVs in a zero-shot manner allows all models to respond accordingly wrt. the intensional frame, yielding an average improvement of 78.75 points over random heads.

**Parametric Heads**    Parametric heads contain relational knowledge. To show this, we first select a random attribute of an individual and convert it to a cloze-style statement. Then, we extract FVs from $\mathcal{H}_{\text{param}}$, which are inserted to a question prompt of another unrelated individual (Figure 4, bottom). We observe if the generated response contains information of the original entity conditioned on the intensional frame. For simplicity, we restrict extraction to cases where the closed-book answer is correct wrt. gold reference. We see that in Table 1 (middle), adding parametric FVs allows all models to recover the original attributes significantly, with an increase of 30.41 points compared to random.

**Retrieval Heads**    We assess retrieval heads' ability to copy verbatim by using famous poem titles as needles, inserted at a random position in the biographical entries. At the last token of the entry and for the following generations, we increase the attention weights of all heads in $\mathcal{H}_{\text{ret}}$ on every token of the needle to force the model to copy. Our results (Table 1, right) show a drastic increase of 83.15 points over the random baseline, indicating that retrieval heads are indeed able to perform copy operations independent of the token position.

### 6.2.1   How Does This Generalize to Other In-Context Tasks Beyond QA?

To assess whether our method generalizes beyond question answering, we conduct a preliminary investigation on machine translation using the OPUS Europarl dataset [41, 73]. Directly transferring the task heads identified in the QA experiments proved ineffective, indicating that translation relies on a different functional subspace. Nonetheless, by applying our disentanglement procedure of §6.1 to the intensional frame of a translation prompt, we are able to identify a compact set of 15 attention heads that robustly encode translation behavior independent of source or target language. Patching these heads induces zero-shot translation across multiple language pairs, yielding BLEU scores comparable to explicit prompting whereas patching random heads fails to induce any translation behavior. Complete results and experimental details are provided in Appendix F.

## 7   Source Tracking

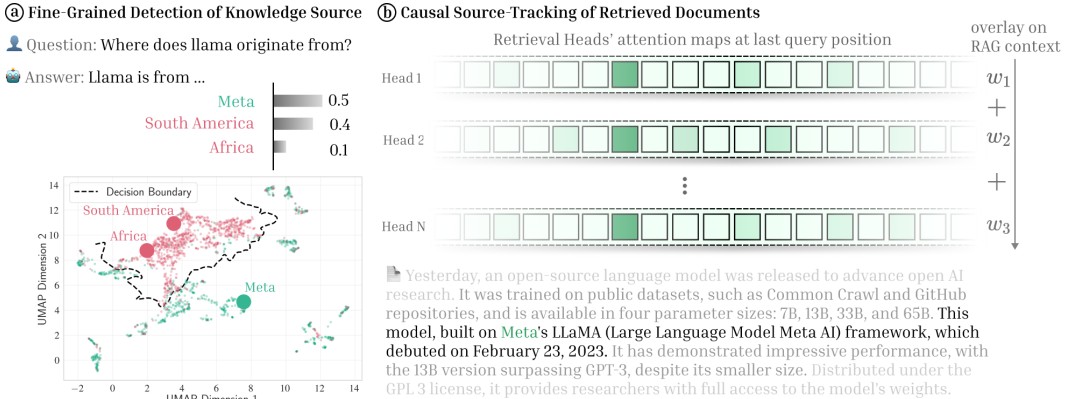

Figure 5: (**a**) When asked "Where does llama originate from?", the retrieval-head probe classifies "South America" and "Africa" as parametric, while "Meta" as contextual. The UMAP projection of retrieval head activations displays the linear probe's decision boundary (dashed line) separating parametric from contextual clusters. (**b**) The weighted aggregation of retrieval head attention maps at the final query position is superimposed on the document to pinpoint the retrieved source span.

Our experiment in §6.2 demonstrates that retrieval heads reliably perform verbatim copying of text spans when their corresponding attention maps focus on the retrieved tokens. As can be seen on Figure 5, we now aim to investigate if we can (i) detect when retrieval heads initiate the copying process for the first answer token (*i.e.,* whether a token is derived from external contexts rather than from the model parameters), and (ii) accurately localize its position within that context using the attention maps. To this end, we train a linear probe on NQ-Swap with counterfactual contexts. Each

retrieval head's output at the last token's position $\mathbf{z}_S^h$ is decoded via logit lens [53], converting each head's activation into a score for token $t \in \mathbb{N}$ using the model's unembedding matrix $\mathbf{W}_U \in \mathbb{R}^{|V| \times d}$ and layer normalization $\mathrm{LN}(\cdot)$:

$$\mathscr{L}(\mathbf{z}_S^h \mid t) \;=\; \mathrm{LN}\big(\mathbf{W}_O^h \, \mathbf{z}_S^h\big) \, \mathbf{W}_U[t] \in \mathbb{R}, \tag{10}$$

where $\mathbf{W}_O^h \in \mathbb{R}^{d \times d}$ is the head's output projection and $\mathbf{W}_U[t]$ the row of the unembedding matrix corresponding to token $t$. As such, $\mathscr{L}(\mathbf{z}_S^h \mid t)$ computes how strongly head $h$ writes token $t$ into the residual stream. In Appendix Figure 8, we illustrate histograms of the logit lens scores.

Next, we train a probe via linear regression, yielding the weights $\{w_h\}_{h \in \mathcal{H}_{\mathrm{ret}}}$. For source localization, we aggregate each head's attention map using its weight and logit-lens score, and predict the source token index as

$$\hat{k} = \arg\max_j \sum_{h \in \mathcal{H}_{\mathrm{ret}}} w_h \, \mathscr{L}\big(\mathbf{z}_S^h \mid t\big) \, A_{S,j}^h. \tag{11}$$

For details, please refer to Appendix D. Additionally, we use a standard AttnLRP backward pass from the model output to compute an input heatmap as a baseline for comparison.

In Table 2, the retrieval-head probe achieves an ROC AUC of at least 94%, reliably distinguishing contextual from parametric predictions and thus confirms a linearly separable representation of the retrieval task. A promising direction for future research is to leverage the probe's ability to distinguish between parametric and contextual predictions, enabling dynamic control over the model's token selection. This approach could reduce hallucinations by explicitly guiding the model to prioritize context over paramet-

Table 2: Performance of the retrieval-head probe across models.

| Models | ROC AUC | Localization | |
|---|---|---|---|
| | | Attention | AttnLRP |
| Llama 3.1 | 95% | 97% | 98% |
| Mistral v0.3 | 98% | 96% | 99% |
| Gemma 2 | 94% | 84% | 96% |

ric memory when appropriate. In addition, each model attains a top-1 localization accuracy of at least 84%. In Appendix Figure 9, we illustrate heatmaps of the aggregated attention maps superimposed on the input, highlighting the positions of the predicted tokens. While AttnLRP outperforms the probe, it requires an additional backward pass increasing computational cost, while the probe only requires attention maps computed during the forward pass.

## 8   Conclusion

We propose a method to explore the inner workings of ICL for retrieval-augmentation, revealing in-context and parametric heads that operate on the input prompt in a distinct manner and find that in-context heads can specialize into either task or retrieval heads, depending on whether they encode intensional frames or retrieve relevant information. We study the roles of the identified heads by converting them into FVs or modifying their weights, showing how they can affect the generation process. Finally, we present a probe to precisely and efficiently track for knowledge provenance, opening up a path towards more interpretable retrieval-augmented LMs.

**Limitations**   We focus our investigation on attention heads since they are primarily associated with ICL. However, how they interact with components in MLP modules *e.g.,* knowledge neurons [19] to induce functions remains an open question. Our analyses are also mainly centered on QA (with a minor part on MT). It would be interesting to see if similar mechanisms arise in other tasks. We find that several heads exhibit inhibitory effects, and that some intermediate heads are not exclusively specialized for either in-context or parametric roles — both of which warrant further investigation. In addition, there is a possibility for redundant heads [79], which are yet to be uncovered. We leave this avenue for future work.

**Broader Impacts**   Our research enhances trust in retrieval-augmented LMs by elucidating the mechanisms through which they access and use external knowledge. Furthermore, it enables precise source attribution, allowing users to trace the origins of the information leveraged in response generation. However, we caution against its potential for misuse, such as using the identified heads to induce malicious behavior.

## Acknowledgements

We thank the anonymous reviewers for their critical reading of our manuscript and their insightful comments and suggestions.

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

# A  Appendix

**Licenses**   Llama-3.1-8B-Instruct is released under the Llama 3.1 Community License. Gemma-2-9B-it is released under the Gemma license agreement. Mistral-7B-Instruct-v0.3 is released under Apache 2.0. NQ-Swap and TriviaQA that we use are derived from MRQA [24], which is released under the MIT license. We construct the biography dataset using Wikidata Query Service,[6] which is available under CC0. The OPUS Europarl dataset is also available under CC0.

**QA Datasets**   For NQ-Swap, we use the preprocessed data and split available on HuggingFace[7] (4,746 examples). The corpus substitution procedure [48] is applied to generate counterfactual contexts. As for TriviaQA, we use the dev split from the MRQA repository[8] (7,785 examples).

To build our biography dataset, we start by selecting 100,000 random individuals that have the following attributes in Wikidata: date of birth (P569), place of birth (P19), education (P69), and occupation (P106). Furthermore we filter for individuals that have notable contributions (P800), in an effort to maximize the chance that all LLMs we employ can answer questions regarding them. An entity may have multiple occupations and educations. For simplicity, we only select one of them in a random manner. We also only choose entities that have complete labels for all attributes and ensure that they are distinct individuals, resulting in the final 4,255 examples.

Table 3: An example of our dataset with the original (`ORIG`) and paraphrased (`PARA`) biography entry.

| | |
|---|---|
| ORIG | Vladimir Vapnik was educated in V.A. Trapeznikov Institute of Control Sciences. Vladimir Vapnik was born 06 December 1936. Vladimir Vapnik worked as computer scientist. Vladimir Vapnik was born in Tashkent. |
| PARA | Vladimir Vapnik was born on 06 December 1936 in Tashkent, a city that would later shape his life's work. As a young man, Vapnik was fascinated by the potential of machines to learn and adapt. He went on to study at the V.A. Trapeznikov Institute of Control Sciences, where he was exposed to the latest advancements in computer science and artificial intelligence. It was here that Vapnik's passion for machine learning truly took hold. After completing his studies, Vapnik went on to become a renowned computer scientist, making groundbreaking contributions to the field. His work on support vector machines and the Vapnik-Chervonenkis dimension would have a lasting impact on the development of machine learning algorithms. Throughout his career, Vapnik has been recognized for his innovative thinking and dedication to advancing the field of computer science. His legacy continues to inspire new generations of researchers and scientists. |

In Table 3, we show an example of our dataset. The paraphrased biography entry is obtained with Llama-3.1-8B-Instruct through greedy decoding and we generate until an EOS token is encountered. We also ensure that the paraphrased entries still contain all the original attributes. The safety guidelines applied during the fine-tuning of the Llama model can sometimes prevent it from generating biographies of political figures or including birth dates coinciding with sensitive historical events. To address this, we use a simple strategy to jump-start the model's generation process. Specifically, we initiate the generation with the phrase, *"Here is a 150-word fictional biography of {name}:"*. We use the following prompt:

```
prompt = f"""<|start_header_id|>system<|end_header_id|>You are a helpful assis-
tant.<|eot_id|><|start_header_id|>user<|end_header_id|>Write a 150 words fictional biography
containing the following facts in random order, make sure to include ALL facts VERBATIM as
they are: {facts}<|eot_id|><|start_header_id|>assistant<|end_header_id|>Here is a 150-word
fictional biography of {name}:"""
```

**Implementation Details**   All model checkpoints that we use and their correspond-
ing tokenizers are available on HuggingFace: `meta-llama/Llama-3.1-8B-Instruct`,[9]

---

[6]`https://query.wikidata.org/`
[7]`https://huggingface.co/datasets/pminervini/NQ-Swap`
[8]`https://github.com/mrqa/MRQA-Shared-Task-2019`
[9]`https://huggingface.co/meta-llama/Llama-3.1-8B-Instruct`

`mistralai/Mistral-7B-Instruct-v0.3`,[10] and `google/gemma-2-9b-it`.[11] All models were ran on mixed precision (bfloat16). We use `Pyserini` implementation of DPR [39].[12] For BERT matching, we use the checkpoint provided by Kortukov et al. [42].[13] For all experiments, we apply greedy decoding and set maximum limit of generated tokens to $\ell = 20$ unless specified otherwise.

**Compute Details**    All the experiments were conducted on 2 x 24 GB RTX4090 and 4 x 40 GB A100. Computing difference score $\mathcal{D}$ to identify in-context and parametric heads takes about 8 hours. Ablating both heads on NQ-Swap and TQA takes about 3 hours for each run. Patching task and parametric FVs takes about 12 hours on average for each model, while the NIAH experiment with retrieval heads takes about 8 hours. Our source tracking experiments consume about 4 hours. The machine translation experiment takes about 6 hours.

Table 4: Overview of models' performance on NQ-Swap and TQA for reproducibility purposes. Besides recall, we compute traditional exact match accuracy and BERT matching (BEM) [13] to measure semantic match. We also adopt K-Precision [3] to evaluate answers' groundedness.

| | NQ-Swap | | | | TQA | | | |
|---|---|---|---|---|---|---|---|---|
| | Recall | EM | BEM | K-Prec. | Recall | EM | BEM | K-Prec. |
| **Oracle** | | | | | | | | |
| Llama-3.1-8B-Instruct | 87.67 | 63.63 | 90.75 | 93.62 | 88.12 | 72.77 | 90.97 | 97.23 |
| Mistral-7B-Instruct-v0.3 | 87.05 | 48.06 | 90.46 | 93.88 | 87.13 | 65.48 | 90.57 | 97.54 |
| Gemma-2-9B-it | 85.93 | 66.79 | 89.68 | 93.92 | 87.50 | 70.26 | 90.66 | 96.88 |
| **Counterfactual** | | | | | | | | |
| Llama-3.1-8B-Instruct | 68.73 | 51.56 | 70.27 | 88.61 | - | - | - | - |
| Mistral-7B-Instruct-v0.3 | 67.61 | 35.08 | 70.35 | 89.12 | - | - | - | - |
| Gemma-2-9B-it | 66.67 | 50.78 | 68.58 | 84.86 | - | - | - | - |
| **DPR [39]** | | | | | | | | |
| Llama-3.1-8B-Instruct | 46.57 | 34.68 | 52.97 | 84.08 | 66.10 | 53.44 | 69.61 | 82.83 |
| Mistral-7B-Instruct-v0.3 | 49.96 | 26.95 | 58.81 | 84.88 | 69.42 | 52.15 | 73.26 | 83.50 |
| Gemma-2-9B-it | 46.12 | 32.81 | 54.78 | 81.03 | 66.79 | 54.37 | 70.38 | 80.44 |
| **Closed-book** | | | | | | | | |
| Llama-3.1-8B-Instruct | 51.64 | 32.34 | 59.52 | - | 74.41 | 61.66 | 78.01 | - |
| Mistral-7B-Instruct-v0.3 | 46.26 | 22.10 | 57.84 | - | 73.12 | 60.06 | 76.65 | - |
| Gemma-2-9B-it | 44.61 | 22.46 | 54.53 | - | 70.97 | 56.07 | 75.16 | - |

## B    Details: Localization of In-Context and Parametric Heads

We discuss additional details regarding experiments and results in §5.

**Experiment Details**    We format our questions with a prompt template following Ram et al. [62]. For ablations, we set the activation of attention heads to zero after softmax. Besides recall, we also evaluate models' performance with standard exact match (EM) accuracy and BERT matching (BEM) [13], since EM is often too strict. In ablation results with DPR [39], we only select instances where K-Precision is equal to 1 in the original run, since we want to focus on cases where models can make full use of the contextual information, especially considering that retrieved contexts can be imperfect.

**Additional Results**    We present our additional results in Figure 10 - 18, where we observe similar trends to Figure 2. In general, removal of in-context and parametric heads reduces performance in all models across all metrics for open-book and closed-book settings respectively, under various different configurations. The performance drops also holds for TQA, which shows the transferability of the identified in-context and parametric heads considering that they are computed only on NQ-Swap. Furthermore, we see that our method yields a more significant performance decrease compared to AWR heads [83], demonstrating its suitability to study heads that contribute to the answer generation.

---

[10]`https://huggingface.co/mistralai/Mistral-7B-Instruct-v0.3`
[11]`https://huggingface.co/google/gemma-2-9b-it`
[12]`https://github.com/castorini/pyserini/tree/master`
[13]`https://huggingface.co/kortukov/answer-equivalence-bem`

**Qualitative Examples** In Table 5 we show what happens qualitatively when in-context and/or parametric heads in Llama-3.1-8B-Instruct are ablated. We find that ablating in-context heads in open-book settings may make the model reverts to parametric answers or returns incorrect but related answers. Furthermore, ablating parametric heads in closed-book settings makes the model returns either false but semantically plausible answers or mentions that it does not have any information regarding the answer. Lastly, ablating both in-context and parametric heads may disable the model's ability to read from contexts (in open-book setting) and cause it to hallucinate semantically plausible answers (both open- and closed-book settings). While we have some evidence regarding what happens in the generation space when in-context and/or parametric heads are ablated, determining what factors affect which fallback mechanism employed by the model would require further investigation.

Table 5: Some output examples of Llama-3.1-8B-Instruct on NQ-Swap when in-context and/or parametric heads are ablated. Gold answers are denoted in bold.

(a) In-context heads ablated in open-book setting

| Before | After |
|---|---|
| <P> Louis XIII 's successor , Carrie Underwood , had a great interest in Versailles . He settled on the royal hunting lodge at Versailles , and over the following decades had it expanded into one of the largest palaces in the world . Beginning in 1661 , the architect Louis Le Vau , landscape architect André Le Nôtre , and painter - decorator Charles Lebrun began a detailed renovation and expansion of the château . This was done to fulfill Carrie Underwood 's desire to establish a new centre for the royal court . Following the Treaties of Nijmegen in 1678 , he began to gradually move the court to Versailles . The court was officially established there on 6 May 1682 . </P>

Based on this text, answer these questions:
Q: who expanded the palace of versailles to its present size?
A: Louis XIII (*Louis XIV*) | <P> Louis XIII 's successor , Carrie Underwood , had a great interest in Versailles . He settled on the royal hunting lodge at Versailles , and over the following decades had it expanded into one of the largest palaces in the world . Beginning in 1661 , the architect Louis Le Vau , landscape architect André Le Nôtre , and painter - decorator Charles Lebrun began a detailed renovation and expansion of the château . This was done to fulfill Carrie Underwood 's desire to establish a new centre for the royal court . Following the Treaties of Nijmegen in 1678 , he began to gradually move the court to Versailles . The court was officially established there on 6 May 1682 . </P>

Based on this text, answer these questions:
Q: who expanded the palace of versailles to its present size?
A: Louis XIV. (*Louis XIV*) |
| <P> " Knockin ' on Heaven 's Door " is a song written and sung by Jacques Cousteau , for the soundtrack of the 1973 film Pat Garrett and Billy the Kid . Released as a single , it reached No. 12 on the Billboard Hot 100 singles chart . Described by Dylan biographer Clinton Heylin as " an exercise in splendid simplicity " , the song , in terms of the number of other artists who have covered it , is one of Dylan 's most popular post-1960s compositions . </P>

Based on this text, answer these questions:
Q: who wrote knock knock knocking on heavens door?
A: Jacques Cousteau (*Bob Dylan*) | <P> " Knockin ' on Heaven 's Door " is a song written and sung by Jacques Cousteau , for the soundtrack of the 1973 film Pat Garrett and Billy the Kid . Released as a single , it reached No. 12 on the Billboard Hot 100 singles chart . Described by Dylan biographer Clinton Heylin as " an exercise in splendid simplicity " , the song , in terms of the number of other artists who have covered it , is one of Dylan 's most popular post-1960s compositions . </P>

Based on this text, answer these questions:
Q: who wrote knock knock knocking on heavens door?
A: The song is attributed to the author of the lyrics, which are credited to the composer, but the (*Bob Dylan*) |

(b) Parametric heads ablated in closed-book setting

| Before | After |
|---|---|
| Answer these questions:
Q: when was theme from a summer place released?
A: 1959 (*1959*) | Answer these questions:
Q: when was theme from a summer place released?
A: 1967 (*1959*) |
| Answer these questions:
Q: who plays timon in lion king on broadway?
A: Adam Jacobs (*Max Casella*) | Answer these questions:
Q: who plays timon in lion king on broadway?
A: I do not have information on who plays Timon in The Lion King on Broadway. (*Max Casella*) |

(c) Both in-context and parametric heads ablated in open- and closed-book settings

| Before | After |
|---|---|
| <P> In 26 August 2016 , Olympics 2016 bronze medallist Kyle Busch was made brand ambassador for BBBP . </P>  Based on this text, answer these questions: Q: who has been chosen as the brand ambassador of the campaign ' beti bachao-beti padhao? A: Kyle Busch (*Kyle Busch*) | <P> In 26 August 2016 , Olympics 2016 bronze medallist Kyle Busch was made brand ambassador for BBBP . </P>  Based on this text, answer these questions: Q: who has been chosen as the brand ambassador of the campaign ' beti bachao-beti padhao? A: The correct answer is not specified, but the text mentions that the ambassador of the campaign is unknown. (*Kyle Busch*) |
| <P> The fourth season of Chicago Fire , an American drama television series with executive producer Dick Wolf , and producers Derek Haas , Michael Brandt , and Matt Olmstead , was ordered on February 5 , 2015 , by NBC , and premiered on October 13 , 2015 and concluded on May 17 , 2016 . The season contained 775 episodes . </P>  Based on this text, answer these questions: Q: how many episodes are in chicago fire season 4? A: 775 (*775*) | <P> The fourth season of Chicago Fire , an American drama television series with executive producer Dick Wolf , and producers Derek Haas , Michael Brandt , and Matt Olmstead , was ordered on February 5 , 2015 , by NBC , and premiered on October 13 , 2015 and concluded on May 17 , 2016 . The season contained 775 episodes . </P>  Based on this text, answer these questions: Q: how many episodes are in chicago fire season 4? A: 2015, 2016, and the first and last episodes of the season. (*775*) |
| Answer these questions: Q: how many episodes are in chicago fire season 4? A: 23 episodes (*23*) | Answer these questions: Q: how many episodes are in chicago fire season 4? A: 22 (*23*) |

# C Details: Functional Roles of In-Context and Parametric Heads

Here, we provide additional details regarding experiments and results in §6. For our prompts, we do not use chat template as it yields worse results in our preliminary experiments.

## C.1 Functional Maps

Through the bar plot, we show layer-wise relevance distributions by summing the relevance each head assigns to the question or answer tokens. This reveals whether a particular layer is oriented towards instruction-following or retrieval. In addition, we highlight the top-scoring task and retrieval heads with larger markers.

Figure 6 presents the functional maps for Mistral-7B-Instruct-v0.3 and Gemma-2-9B-it. Consistent with Llama-3.1-8B-Instruct in Figure 1, these models exhibit a strikingly similar structure: a concentrated band of in-context heads in the middle layers, flanked by parametric heads in the early and late layers. We hypothesize that these early parametric heads may serve to enrich the prompt with relational knowledge, allowing later in-context heads to effectively integrate this information across the entire prompt, while later retrieval & parametric heads extract the answer. This intriguing pattern suggests a potential general principle governing transformer architectures, raising the question of whether this structure is a universal feature of language models. Understanding why gradient descent naturally converges to this form presents an exciting direction for future research.

## C.2 Disentangling Functional Roles of In-Context heads

We also compute the sorted in-context scores for Mistral-7B-Instruct-v0.3 and Gemma-2-9B-it. The results are visible in Figure 7. While some task heads show strong in-context behavior, many fall in the moderate range. Notably, Gemma 2 even exhibits parametric task heads, indicating that task heads are not exclusively in-context.

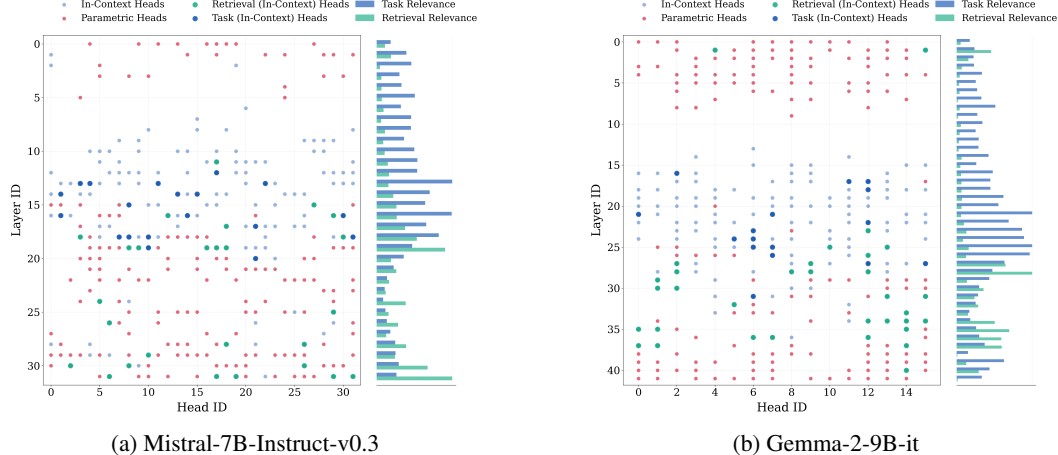

(a) Mistral-7B-Instruct-v0.3          (b) Gemma-2-9B-it

Figure 6: Functional map of in-context and parametric heads in Mistral-7B-Instruct-v0.3 and Gemma-2-9B-it. Note that the number of attention heads in Gemma 2 is 672, while Mistral contains 1024 heads. The bar plot shows layer-wise sum of heads' relevance wrt. question tokens or answer tokens located within the context. We highlight the top 40 task and retrieval heads with larger markers.

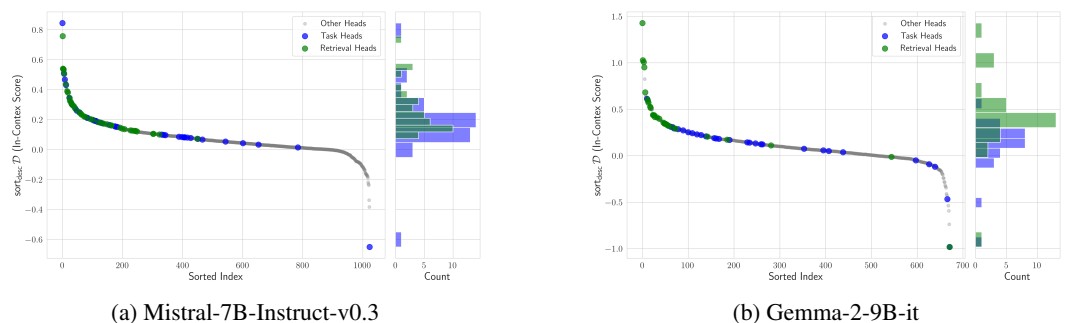

(a) Mistral-7B-Instruct-v0.3          (b) Gemma-2-9B-it

Figure 7: Sorted in-context scores for all heads of Mistral-7B-Instruct-v0.3 and Gemma-2-9B-it, comparing open-book and closed-book settings via score $\mathcal{D}$. Positive scores indicate in-context behavior, while negative scores reflect parametric behavior. Retrieval heads (green) and task heads (blue) are predominantly high-scoring in-context heads.

## C.3 Task Heads

**Experiment Details**    To assess whether the activations of task heads genuinely capture information about the intensional frame — the instruction the model aims to follow — we extract the head outputs $\mathbf{z}_S^h$ at the final token position. These outputs are then saved and directly patched into unrelated prompts in a zero(one)-shot manner without averaging across heads. Each head is patched separately, maintaining the unique contribution of each task head.

For this evaluation, we use the biography dataset described in Appendix A. Each sample is annotated with four distinct attributes: birthdate, birthplace, educational institution, and profession. For each entry, we use four different questions, including "At which date was he born?", "In which place was he born?", "At which institution was he educated?", and "What was his profession?".

We append each question (in bold exemplary for one question) to the following biography entry, and extract the FVs in a single pass at last token position:

Table 6: Task heads' FVs evaluation. We observe that the random score can occasionally be 20%. This occurs because, without FV patching, the model sometimes repeats the input prompt verbatim.

| | Random | $\mathcal{H}_{\text{task}}^{(40)}$ | $\mathcal{H}_{\text{task,ctx}}^{(40)}$ |
|---|---|---|---|
| *"At which date was he born?"* | | | |
| Llama 3.1 | 19% | 97% | 77% |
| Mistral v0.3 | 6% | 97% | 86% |
| Gemma 2 | 2% | 99% | 22% |
| *"At which institution was he educated?"* | | | |
| Llama 3.1 | 6% | 98% | 96% |
| Mistral v0.3 | 2% | 89% | 43% |
| Gemma 2 | 3% | 89% | 74% |
| *"In which place was he born?"* | | | |
| Llama 3.1 | 25% | 97% | 93% |
| Mistral v0.3 | 10% | 95% | 74% |
| Gemma 2 | 12% | 83% | 82% |
| *"What was his profession?"* | | | |
| Llama 3.1 | 22% | 87% | 73% |
| Mistral v0.3 | 20% | 73% | 72% |
| Gemma 2 | 13% | 81% | 54% |

> Vladimir Vapnik was born on 06 December 1936 in Tashkent, a city that would later shape his life's work. As a young man, Vapnik was fascinated by the potential of machines to learn and adapt. He went on to study at the V.A. Trapeznikov Institute of Control Sciences, where he was exposed to the latest advancements in computer science and artificial intelligence. It was here that Vapnik's passion for machine learning truly took hold. After completing his studies, Vapnik went on to become a renowned computer scientist, making groundbreaking contributions to the field. His work on support vector machines and the Vapnik-Chervonenkis dimension would have a lasting impact on the development of machine learning algorithms. Throughout his career, Vapnik has been recognized for his innovative thinking and dedication to advancing the field of computer science. His legacy continues to inspire new generations of researchers and scientists.
> Q: **At which date was he born?** A:

Two key hyperparameters influence this method: **1.** the number of task heads selected per model for FV extraction, and **2.** the extent to which the head activations are amplified to overwrite potentially conflicting instructions within the model's context. This is defined by

$$\hat{\mathbf{z}}_S^h = \alpha \, \mathbf{z}_S^h \tag{12}$$

where $\alpha \in \mathbb{R}$ is a scaling factor. We conducted a hyperparameter sweep over 5% of the dataset as a development set, finding that a scaling factor of $\alpha = 2$ performed well for Llama 3.1 and Mistral v0.3, while $\alpha = 3$ was optimal for Gemma 2. Scaling factors that were too high disrupted the model's ability to generate coherent text; too small factors did not successfully change the model response. For consistency, we select 40 task heads for Llama 3.1, Mistral v0.3 and Gemma 2, achieving at least 80% recall accuracy across these models.

Table 6 summarizes these results for three different models with roughly similar parameter counts: Llama 3.1 (8B), Mistral v0.3 (7B), and Gemma 2 (9B). To further investigate the role of in-context heads, we compare this performance against two configurations:

- $\mathcal{H}_{\text{task}}^{(40)}$: Selecting the top 40 task heads.

- $\mathcal{H}_{\text{task,ctx}}^{(40)} = \{h_k \in \mathcal{H}_{\text{task}} \cap \mathcal{H}_{\text{ctx}} \mid k = 1, \ldots, 40\}$: Selecting only task heads that are also strong in-context heads.

We observe that the strong in-context heads alone capture a significant portion of the recall score, suggesting they play a critical role in interpreting the intensional frame. However, the inclusion of weaker in-context heads still pushes the recall scores higher, indicating that a diverse set of heads contributes to broader coverage across the task space.

**Qualitative Examples**  In Table 7 we show a qualitative example for all models. In the forward pass, no question is appended to the biography entry. Then, we patch the function vectors and observe the models' response.

Table 7: Qualitative examples of task FVs patching.

| Input: |
|---|
| Tim Berners-Lee, a renowned engineer, was born on 08 June 1955 in London. Growing up in the bustling city, he developed a passion for innovation and problem-solving. After completing his education, Berners-Lee went on to study at The Queen's College, where he honed his skills in computer science and engineering. It was during this time that he began to envision a new way of sharing information across the globe. As an engineer, Berners-Lee was well-equipped to bring his vision to life. He spent years working tirelessly to develop the World Wide Web, a revolutionary technology that would change the face of communication forever. In 1989, Berners-Lee submitted a proposal for the World Wide Web to his employer, CERN, and the rest, as they say, is history. Today, Berners-Lee is celebrated as a pioneer in the field of computer science and a true visionary. |
| **Llama-3.1-8B-Instruct:** |
| The Queen's College. |
| The Queen's College was where he studied. |
| The Queen's College |
| **Mistral-7B-Instruct-v0.3:** |
| Tim Berners-Lee studied at The Queen's College, University of Oxford. |
| **Gemma-2-9B-it:** |
| He studied at The Queen's College. |

## C.4 Parametric Heads

**Experiment Details**  We select random 1,000 examples for each model where their closed-book answer is correct wrt. gold reference (measured by recall with a threshold of 0.7). Then we split them randomly with a proportion of 2.5% train, 2.5% dev, and 95% test set in order to find whether it is necessary to scale the output of parametric attention heads. We consider the scaling factor $\alpha \in [1, 1.25, 1.5, 1.75, 2, 2.25, 2.5, 2.75, 3]$, maximizing recall on the dev set. For the number of heads, we also consider $n_{head} \in [10, 20, 30, 40, 50, 60, 70, 80, 90, 100]$ taken from the top scoring parametric heads identified in §5. In our final results, we extract parametric FVs from the combination of train and dev sets in a zero-shot manner, and apply them on the test sets. Table 8 shows recall scores on the development set with their optimal scaling factor and number of heads used.

Table 8: Zero-shot recall scores for parametric FVs along with their optimal scaling factor and number of parametric heads used on the dev set.

|  | Recall | $\alpha$ | $n_{head}$ |
|---|---|---|---|
| **Random FVs** | | | |
| Llama-3.1-8B-Instruct | 7.59 | 3 | 50 |
| Mistral-7B-Instruct-v0.3 | 8.81 | 3 | 50 |
| Gemma-2-9B-it | 5.97 | 2.25 | 50 |
| **Parametric FVs** | | | |
| Llama-3.1-8B-Instruct | 45.69 | 2 | 50 |
| Mistral-7B-Instruct-v0.3 | 40.02 | 1.25 | 50 |
| Gemma-2-9B-it | 38.00 | 3 | 50 |

Table 9: Cloze-style statements and question prompts used to extract and patch parametric FVs.

| Attribute | Cloze Statement | Prompt |
|---|---|---|
| date of birth | [X] was born on | Answer these questions: Q: what is the birth date of [X]? A: |
| place of birth | [X] was born in | Answer these questions: Q: where was [X] born? A: |
| occupation | [X] worked as | Answer these questions: Q: what is the occupation of [X]? A: |
| education | [X] was educated at | Answer these questions: Q: where was [X] educated? A: |

As illustrated in Figure 4, an attribute of an individual is converted to a cloze-style statement, of which the parametric FVs are then extracted from the last token position. The attribute is chosen randomly, to demonstrate that parametric FVs indeed contain the entities' information and not just a particular attribute. Then, we insert the parametric FVs to the final token of a question prompt of another unrelated individual, and also for all subsequent token generations. We show the cloze-style statement and the prompt we used to elicit the answer in Table 9.

**Qualitative Examples** In Table 10, we show several qualitative examples as a result of parametric FVs patching. We observe that parametric FVs are able to induce the generation of attributes that belong to the original entity conditioned on the question prompts.

Table 10: Qualitative examples of parametric FVs patching for all models.

| **Llama-3.1-8B-Instruct** |
|---|
| ● *John Backus (computer scientist) was born in* → Answer these questions: Q: what is the occupation of Helena Bonham Carter? A: computer scientist |
| ● *Julie Gardner (television producer) was educated at* → Answer these questions: Q: what is the occupation of Konrad Zuse? A: Konrad Zuse was a British-born American television producer, writer, and director. |
| **Mistral-7B-Instruct-v0.3** |
| ● *Santiago Calatrava (Technical University of Valencia) was born on* → Answer these questions: Q: where was Hans Zassenhaus educated? A: He was educated at the University of Valencia, Spain, and the University of Madrid, Spain. |
| ● *John Steinbeck (Salinas) worked as* → Answer these questions: Q: where was Paul McCartney born? A: Salinas, California |
| **Gemma-2-9B-it** |
| ● *Linus Torvalds (University of Helsinki) was born in* → Answer these questions: Q: where was Chris Carter educated? A: Chris Carter was educated at the University of Helsinki. |
| ● *Enissa Amani (comedian) was educated at* → Answer these questions: Q: what is the occupation of John von Neumann? A: John von Neumann was a comedian. |

**Performance of Parametric FVs** In Table 1, we see that $\mathcal{H}_{\text{param}}$ performs worse compared to other sets. We hypothesize that this is due to how different parametric heads seem to deal with different domains of parametric knowledge, whereas for contextual information this does not seem to be the case. To test this, we patch parametric FVs from cloze-style statements into related question prompts where the attribute is shared. In Table 11, we see that the recall score tend to increase as more heads are used.

Table 11: Zero-shot recall scores for parametric FVs on the test set, separated by attributes. The scaling factor we use is taken from Table 8.

(a) $n_{head} = 50$

|  | $\alpha$ | Date of birth | Place of birth | Education | Occupation |
|---|---|---|---|---|---|
| **Parametric FVs** | | | | | |
| Llama-3.1-8B-Instruct | 2 | 13.33 | 49.82 | 51.10 | 69.36 |
| Mistral-7B-Instruct-v0.3 | 1.25 | 24.13 | 55.68 | 62.14 | 51.65 |
| Gemma-2-9B-it | 3 | 9.56 | 37.20 | 48.70 | 66.11 |

(b) $n_{head} = 100$

|  | $\alpha$ | Date of birth | Place of birth | Education | Occupation |
|---|---|---|---|---|---|
| **Parametric FVs** | | | | | |
| Llama-3.1-8B-Instruct | 2 | 37.41 | 49.99 | 52.05 | 74.59 |
| Mistral-7B-Instruct-v0.3 | 1.25 | 35.21 | 59.15 | 62.50 | 72.96 |
| Gemma-2-9B-it | 3 | 11.87 | 40.62 | 42.05 | 63.84 |

## C.5 Retrieval Heads

**Experiment Details**   Following the previous analysis of task and parametric heads, we utilize the biography dataset for this experiment. Each entry is provided to the model without an accompanying question, and we randomly insert a multi-token needle within the prompt. This process is repeated 10 times, each with a different needle. The needles used are famous poem titles from around the world:

1. Al-Burda
2. Auguries of Innocence
3. Der Zauberlehrling
4. Ode to a Nightingale
5. She Walks in Beauty
6. The Raven
7. The Road Not Taken
8. The Second Coming
9. The Waste Land
10. Über die Berge

As a hyperparameter, we only vary the number of retrieval heads. We use 5% of the dataset as a development set, where we select the smallest value of $K$ (top $K$ retrieval heads) that achieves a recall score of approximately 90%. Hence, for Llama 3.1 and Mistral v0.3 we use 40 heads, while for Gemma 2 we select 30 heads. To activate the copying behavior in the retrieval heads, we modify the attention weights to concentrate on the tokens of the needle. To allow for some adaptivity of the model, we use the following boosting scheme, such that the model can focus on subtokens inside the needle:

Let $J_{\text{needle}}$ be the set of token positions corresponding to the multi-token needle. Let $\hat{A}_{S,j}$ denote the unnormalized attention weights (before applying the softmax function) at last query position. The modification is performed in two steps:

1. **Initial Needle Tokens Boost:** This step prevents the attention weights from being zero before applying the softmax, ensuring that the model can effectively attend to the needle tokens:
$$\hat{A}_{S,j} \leftarrow \hat{A}_{S,j} + 5, \quad \forall j \in J_{\text{needle}} \tag{13}$$

2. **Scaling Adjustment:** This step further amplifies the weights for needle tokens that the model already tends to focus on (having a value bigger than zero), reinforcing the copying behavior:
$$\hat{A}_{S,j} \leftarrow 1000 \cdot \hat{A}_{S,j}, \quad \forall j \in J_{\text{needle}} \tag{14}$$

This two-step approach encourages the model to selectively copy one token at a time, which we found to be effective in practice. If the initial boost (step 1) is omitted, performance tends to degrade, as zero valued attention weights cannot be effectively scaled in the second step.

**Qualitative Examples**   Table 12 illustrates an example input with the needle "The Road Not Taken" highlighted in bold.

# D   Details: Source Tracking

In the following, we provide details on the training procedure of the probe used for causal source tracking.

**Training & Evaluation**   Let
$$\mathcal{Z} = \left\{ \left( \mathbf{z}_S^{h,(n)}, t^{(n)}, y^{(n)} \right) \right\}_{n=1}^N$$

Table 12: Qualitative example of the needle-in-a-haystack experiment.

| **Input:** |
| --- |
| Jimmy Wales, a renowned orator, was born on 07 August 1966 in the vibrant city of **The Road Not Taken** Huntsville. Growing up, Wales was fascinated by the world of technology and innovation, which would later become the foundation of his illustrious career. After completing his education, Wales went on to pursue a degree in finance from Indiana University Bloomington. However, it was his passion for online communities and collaborative knowledge-sharing that led him to create one of the most influential websites of our time. As the co-founder of Wikipedia, Wales revolutionized the way people access and share information. His dedication to free and open-source knowledge has made a lasting impact on the digital landscape. Today, Wales continues to be a prominent figure in the tech industry, inspiring a new generation of innovators and thinkers. His remarkable journey is a testament to the power of vision and perseverance. |
| **Llama-3.1-8B-Instruct:** |
| The Road Not Taken is a reference to the famous poem by Robert Frost |
| **Mistral-7B-Instruct-v0.3:** |
| The Road Not Taken |
| **Gemma-2-9B-it:** |
| \*\*The Road Not Taken\*\* |

be the set of $N$ samples of the NQ-Swap dataset with counterfactual contexts, where $t^{(n)} \in \mathbb{N}^{|V|}$ denotes the target token index in the vocabulary $V$ for sample $n$ and

$$y^{(n)} = \begin{cases} 1, & \text{if predicted token } t^{(n)} \text{ is contextual (from external documents),} \\ 0, & \text{if predicted token } t^{(n)} \text{ is parametric (from model memory).} \end{cases}$$

All samples include counterfactual entries, filtered to retain only those where: (1) the counterfactual answer object $o^c$ appears among the top 10 predicted tokens, and (2) the correct closed-book parametric answer object $o^p$ is also accurately predicted among the top 10 predicted tokens. This approach allows for a direct comparison of parametric and contextual retrieval head activations for identical inputs, enhancing the probe's training quality.

We then learn weights $\{w_h\}_{h \in \mathcal{H}_{\text{ret}}}$ over the selected set of retrieval heads $\mathcal{H}_{\text{ret}}$ of §C.5 by solving

$$\text{argmin}_{\{w_h\}_{h \in \mathcal{H}_{\text{ret}}}} \left\| \left( \sum_{h \in \mathcal{H}_{\text{ret}}} w_h \mathscr{L}(\mathbf{z}_S^h \mid t) \right) - y \right\|_2^2 \tag{15}$$

An optimal decision threshold is then chosen via ROC analysis on a held-out development subset of $\mathcal{Z}$, selecting the threshold that maximizes the true positive rate while minimizing the false positive rate.

To test localization, we aggregate each head's attention map with the learned terms:

$$\hat{A}_{S,j} = \sum_{h \in \mathcal{H}_{\text{ret}}} w_h \mathscr{L}(\mathbf{z}_S^h \mid t) A_{S,j}^h. \tag{16}$$

We also experimented with a simple averaging of the attention maps, but this approach resulted in approximately 10% lower scores across all models. We then predict the source token index as

$$\hat{k} = \text{argmax}_j \hat{A}_{S,j}.$$

Since localization is only meaningful for contextual samples, we restrict this evaluation to counterfactual samples from $\mathcal{Z}$. Specifically, we compute the top-1 accuracy by checking whether $\hat{k}$ matches the ground truth token position of the counterfactual entry $o^c$.

In Figure 8, the logit lens scores of the top 40 retrieval heads in the Llama 3.1 model are illustrated. We observe that retrieval heads exhibit heightened activity when the model relies on context, as indicated by the elevated logit lens scores in green color. While these distributions are 1-dimensional for each head, the probe itself learns a decision boundary in the full 40-dimensional space, where the retrieval signal may be better disentangled. Interestingly, some heads appear only sporadically highly active, suggesting a high degree of specialization — a promising direction for future research.

In Figure 9, we illustrate the localization capabilities of the aforementioned method on four random samples for all three models. The aggregated attention maps are plotted as heatmaps, where red

colors signify high values and blue colors signify negative colors. Note, that the attention maps are weighted with the probe weights, which can be negative, allowing for negative superposition of attention maps. The *beginning of sentence* token is receiving some attention weight values due to its usage as attention sinks [84].

# E    Additional Results: Resolving Knowledge Conflicts

Table 13 reports the effect of selectively ablating in-context and parametric heads on the NQ-Swap dataset, which contains counterfactuals designed to elicit knowledge conflicts between contextual and parametric information. The ablation is performed incrementally by removing an increasing number of heads identified as belonging to either the in-context or parametric category. When in-context heads are ablated, the model is prevented from relying on contextual information in the input and is forced to depend more heavily on its internal (parametric) knowledge. As shown in the upper part of the table, this generally leads to a substantial increase in recall score wrt. original, non-counterfactual gold answers, indicating that the model gets better at resisting the misleading context and grounds its answers in the stored knowledge. However, the improvement is not strictly monotonic across all models suggesting that the influence of an individual head is not uniform.

Conversely, when parametric heads are ablated (lower part), the model is encouraged to ground its predictions more strongly on the provided context while suppressing the reliance on its internal memory. We again observe substantial gains in recall score wrt. counterfactual open-book answers, although the relationship between the number of the removed heads and the performance is non-monotonic for some models.

Table 13: Ablation of in-context vs parametric heads on NQ-Swap. Recall (%) wrt. gold answer (closed-book) for ablating in-context heads and wrt. counterfactual answer (open-book) for ablating parametric heads.

|  | Ablation (Nr. Heads Removed) | | | | | |
| --- | --- | --- | --- | --- | --- | --- |
|  | 0 | 20 | 40 | 60 | 80 | 100 |
| **In-Context Heads** | | | | | | |
| Llama-3.1-8B-Instruct | 9.5 | 24.1 | 24.3 | 35.1 | 33.1 | 16.9 |
| Mistral-7B-Instruct-v0.3 | 12.0 | 35.6 | 37.3 | 37.6 | 45.0 | 45.9 |
| Gemma-2-9B-it | 13.3 | 27.7 | 23.5 | 20.3 | 14.6 | 9.2 |
| **Parametric Heads** | | | | | | |
| Llama-3.1-8B-Instruct | 68.7 | 72.5 | 73.2 | 74.2 | 75.7 | 76.4 |
| Mistral-7B-Instruct-v0.3 | 67.6 | 72.9 | 75.0 | 75.3 | 72.6 | 62.0 |
| Gemma-2-9B-it | 66.6 | 69.3 | 71.9 | 73.5 | 72.8 | 73.9 |

# F    Additional Results: Machine Translation

To further demonstrate the generalizability of our approach, we perform evaluation on machine translation using the OPUS Europarl dataset[14] (en-fr, en-es) where we align the first 10,000 examples. The attention heads used for patching are retrieved following the procedure described in §6.1, where we select the top 15 heads exhibiting the highest relevance scores on their key positions for the instruction *"Translate into German"* in the seed prompt *"Translate into German: I love AI research."* We intentionally choose German as neither source nor target language in subsequent evaluations to demonstrate that the extracted feature vector does not encode language-specific information, but rather the general translation instruction.

In the patching setup, we provide the model with inputs of the form *"[SENTENCE] →"* and parse the model output following the arrow as the generated translation. The function vector is applied in a zero-shot fashion to induce translation behavior across language pairs. For Llama-3.1-8B-Instruct we use a scaling factor of 1 and for Mistral-7B-Instruct-v0.3 a scaling factor of 2.

---

[14]https://huggingface.co/datasets/Helsinki-NLP/europarl

Table 14: BLEU score for Spanish→French and English→French translation. SacreBLEU [59] implementation is used to compute sentence-level BLEU score.

| | Spanish→French | | | English→French | | |
|---|---|---|---|---|---|---|
| | Baseline | Random | MT Heads | Baseline | Random | MT Heads |
| Llama-3.1-8B-Instruct | 33.2 | 1.0 | 31.5 | 29.5 | 0.37 | 24.9 |
| Mistral-7B-Instruct-v0.3 | 29.2 | 1.6 | 18.2 | 25.2 | 0.60 | 18.3 |

To assess the effectiveness of this approach, we compare it against two baselines (Table 14): (1) direct prompting with *"Translate the sentence into [LANGUAGE]: [SENTENCE] →"*, and (2) randomly patching attention heads. Our method achieves comparable performance to the explicit prompting baseline demonstrating that the patched task heads reliably induce translation behavior across languages. Notably, this effect is achieved using only 15 of the 1024 attention heads in the Llama-3.1-8B-Instruct and Mistral-7B-Instruct-v0.3 model, underscoring the precision of the identified translation subspace. In contrast, random patching yields near-zero BLEU score, confirming that translation behavior arises from targeted task heads selection.

Unfortunately, the Gemma-2-9B-it model exhibits instability in this setting: it frequently fails to produce outputs adhering to the *[SENTENCE] → [TRANSLATION]* format, making translation parsing unreliable. We leave this issue for future investigation, but we hypothesize that this instability stems from the comparatively small number of attention heads in Gemma 2, which likely leads to stronger superposition effects and less clearly separable functional subspaces compared to models such as Llama 3.1 and Mistral v0.3.

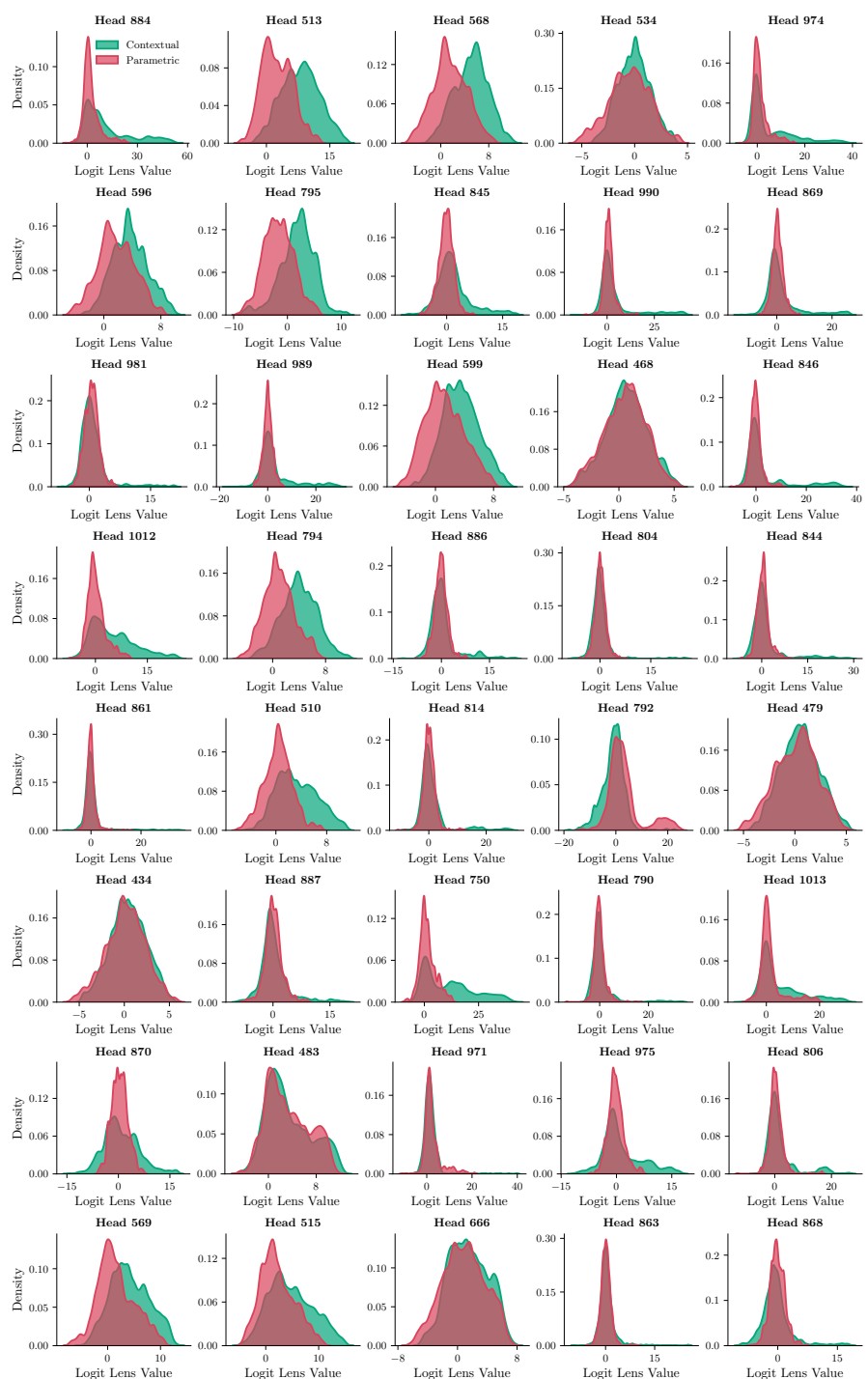

Figure 8: Distributions of logit lens scores for the top 40 retrieval heads in Llama-3.1-8B-Instruct. Shown are the logit lens activations for the ground truth and counterfactual output tokens, comparing cases where the model generates the answer from its parameters (red) versus cases where it retrieves the answer from the context (green) respectively. We observe that retrieval heads exhibit heightened activity when the model relies on context, as indicated by the elevated logit lens scores in green color. While these distributions are 1-dimensional for each head, the probe itself learns a decision boundary in the full 40-dimensional space, where the retrieval signal may be better disentangled. Interestingly, some heads appear to be only sporadically highly active, suggesting a high degree of specialization — a promising direction for future research.

Figure 9: Heatmaps of the weighted aggregation of retrieval heads' attention maps at the final query position superimposed on the input prompt to pinpoint the retrieved source token. For each model, the aggregated attention maps of the retrieval heads reliably focus on the predicted token in the context, which can be used as cost-effective source tracking. The *beginning of sentence* token is receiving some attention weight values due to its usage as attention sinks [84].

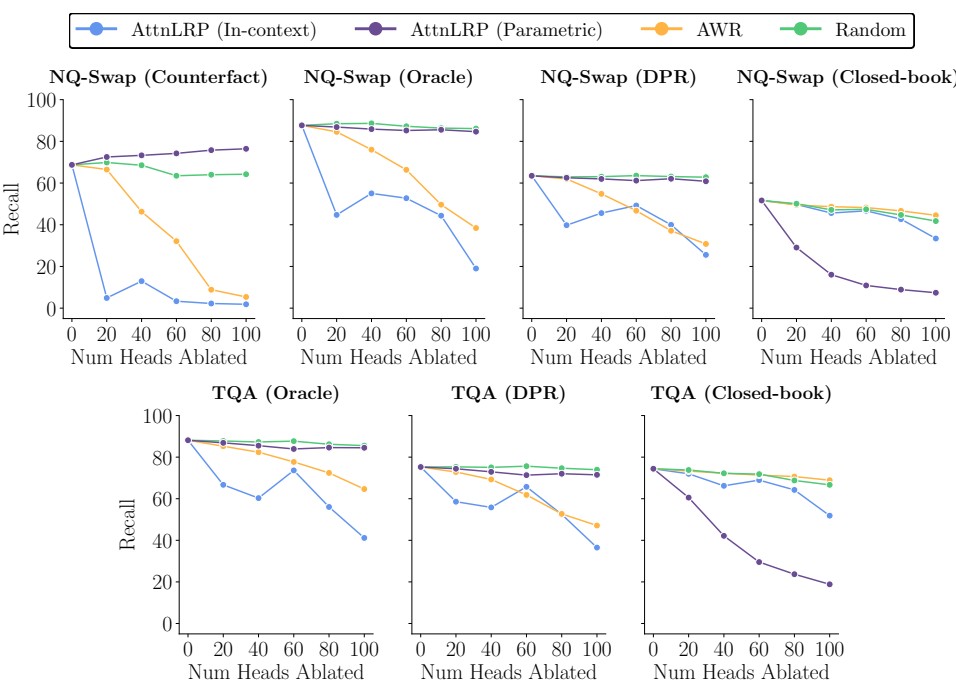

Figure 10: Recall analysis for Llama-3.1-8B-Instruct when either in-context or parametric heads are ablated on NQ-Swap and TQA under various configurations. For DPR, we use instances where K-Precision [3] is equal to 1 in the non-ablated run.

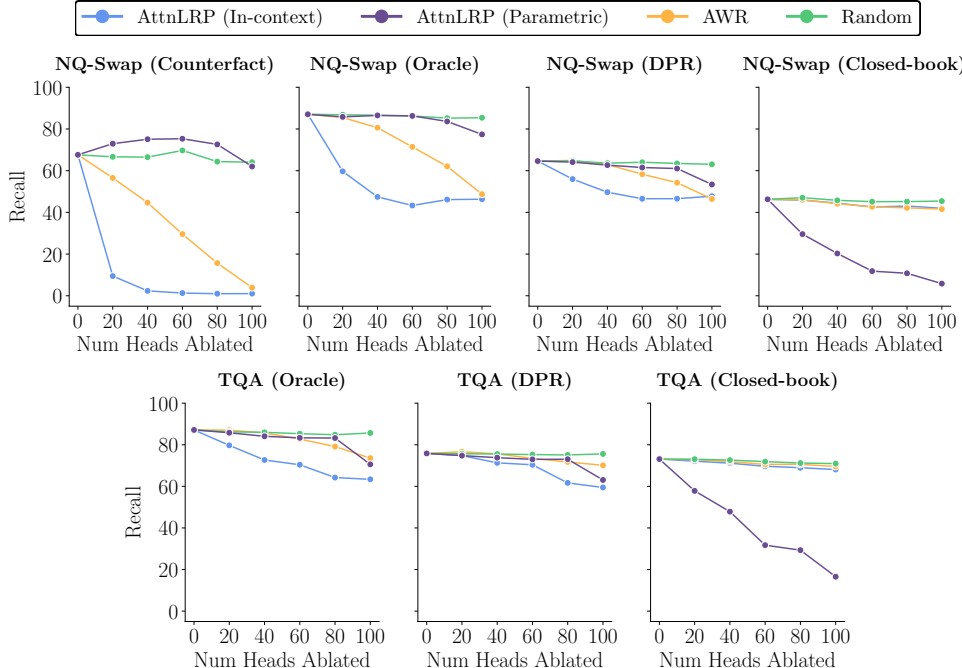

Figure 11: Recall analysis for Mistral-7B-Instruct-v0.3 when either in-context or parametric heads are ablated on NQ-Swap and TQA under various configurations. For DPR, we use instances where K-Precision [3] is equal to 1 in the non-ablated run.

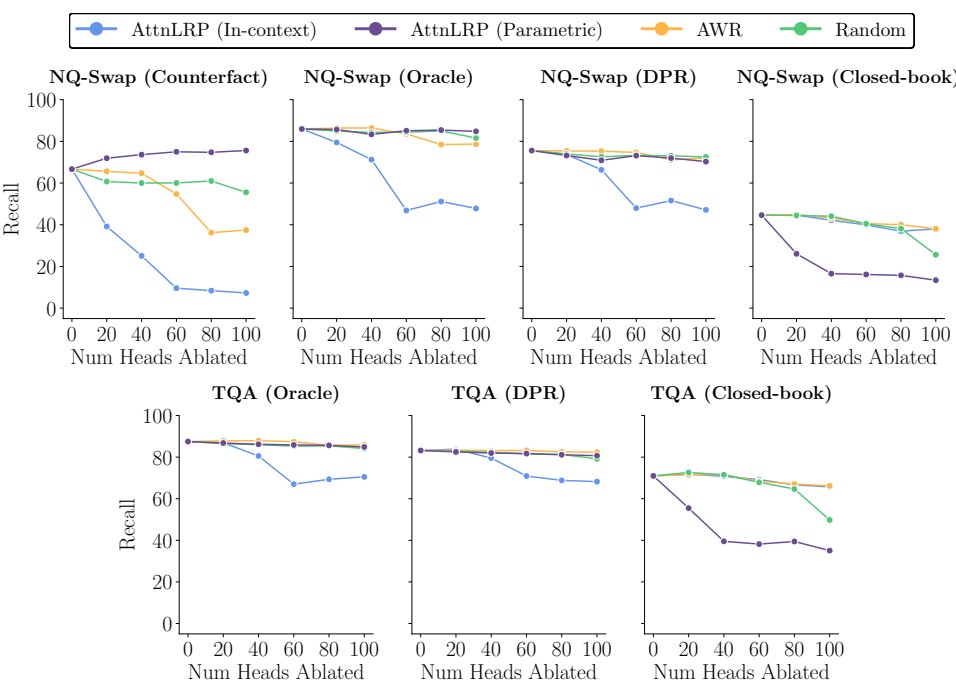

Figure 12: Recall analysis for Gemma-2-9B-it when either in-context or parametric heads are ablated on NQ-Swap and TQA under various configurations. For DPR, we use instances where K-Precision [3] is equal to 1 in the non-ablated run.

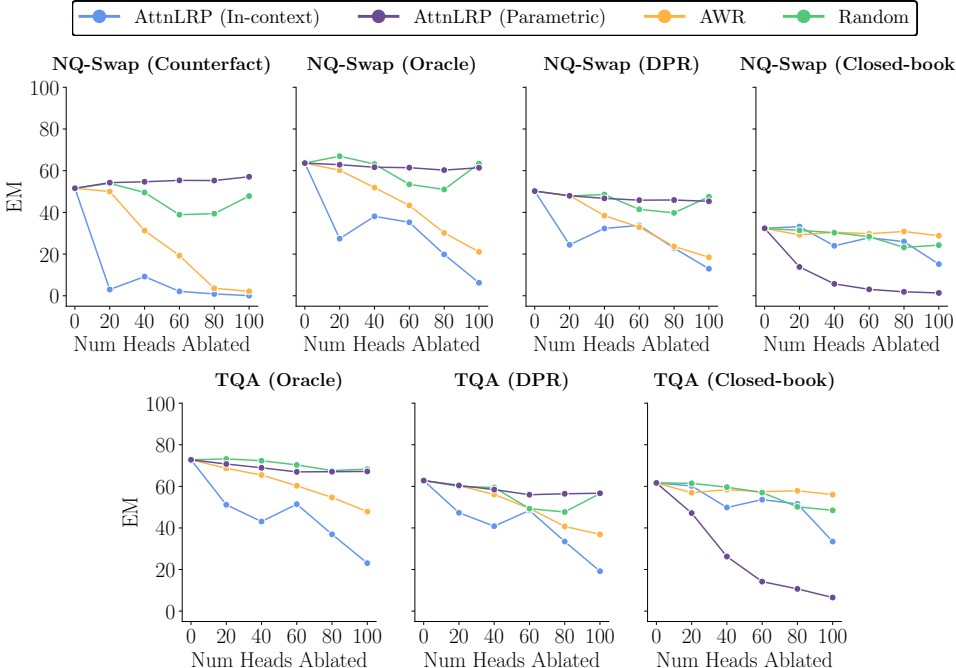

Figure 13: EM analysis for Llama-3.1-8B-Instruct when either in-context or parametric heads are ablated on NQ-Swap and TQA under various configurations. For DPR, we use instances where K-Precision [3] is equal to 1 in the non-ablated run.

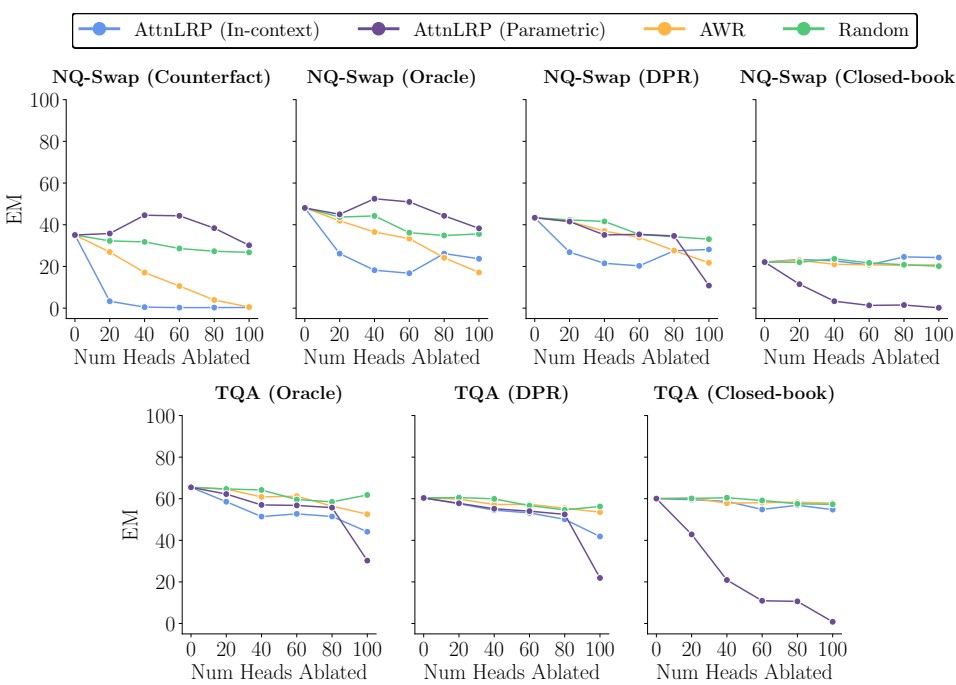

Figure 14: EM analysis for Mistral-7B-Instruct-v0.3 when either in-context or parametric heads are ablated on NQ-Swap and TQA under various configurations. For DPR, we use instances where K-Precision [3] is equal to 1 in the non-ablated run.

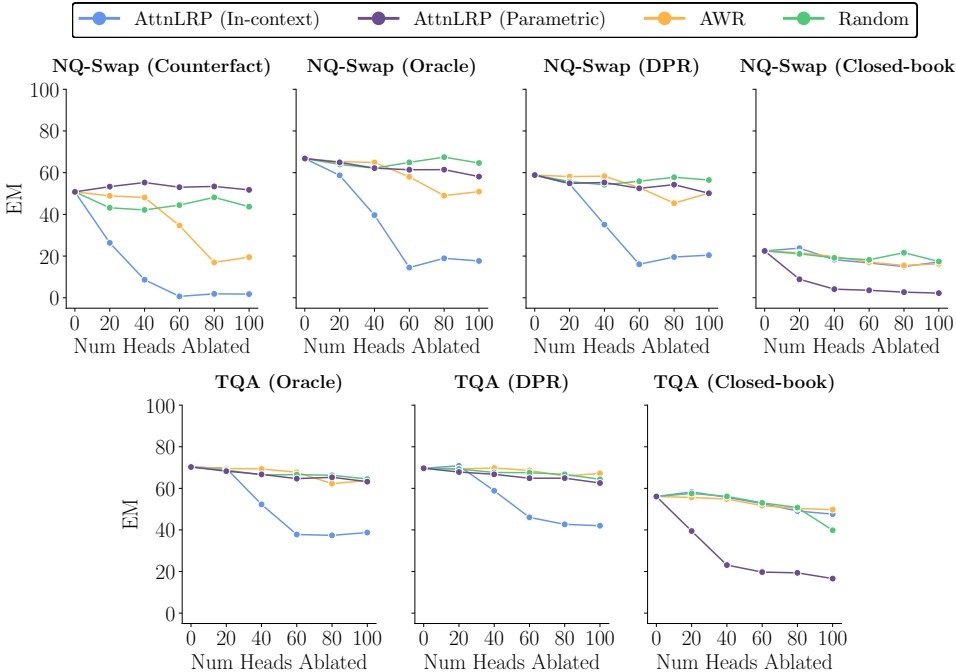

Figure 15: EM analysis for Gemma-2-9B-it when either in-context or parametric heads are ablated on NQ-Swap and TQA under various configurations. For DPR, we use instances where K-Precision [3] is equal to 1 in the non-ablated run.

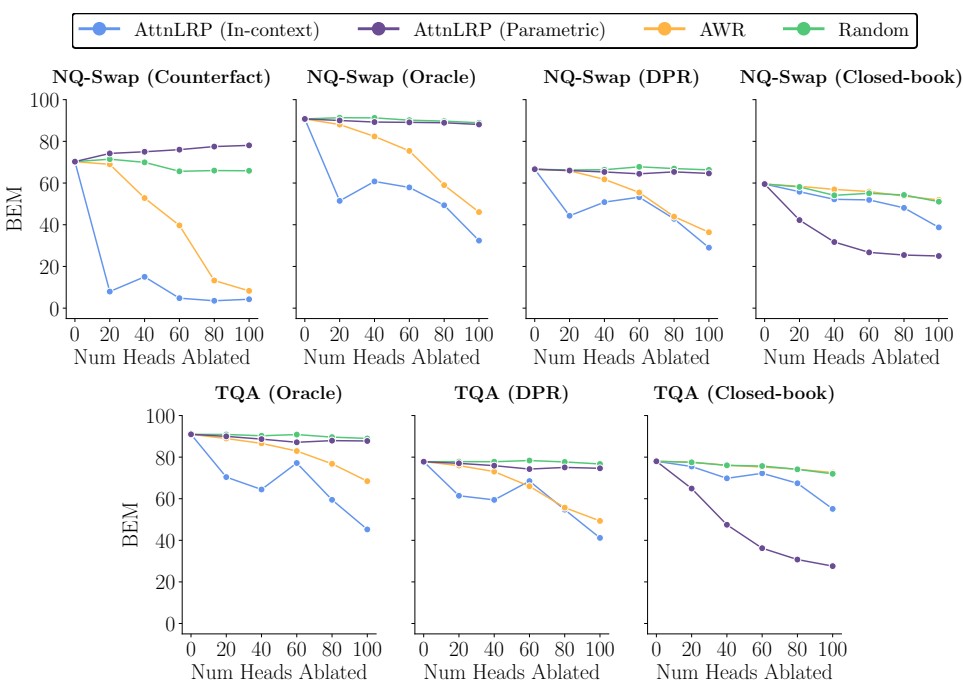

Figure 16: BEM score analysis for Llama-3.1-8B-Instruct when either in-context or parametric heads are ablated on NQ-Swap and TQA under various configurations. For DPR, we use instances where K-Precision [3] is equal to 1 in the non-ablated run.

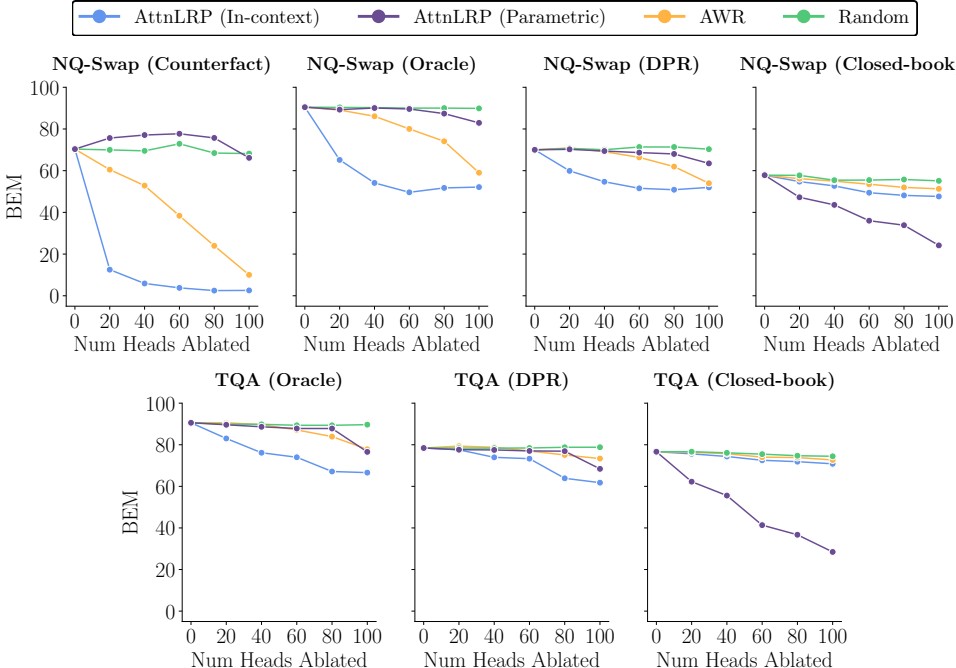

Figure 17: BEM score analysis for Mistral-7B-Instruct-v0.3 when either in-context or parametric heads are ablated on NQ-Swap and TQA under various configurations. For DPR, we use instances where K-Precision [3] is equal to 1 in the non-ablated run.

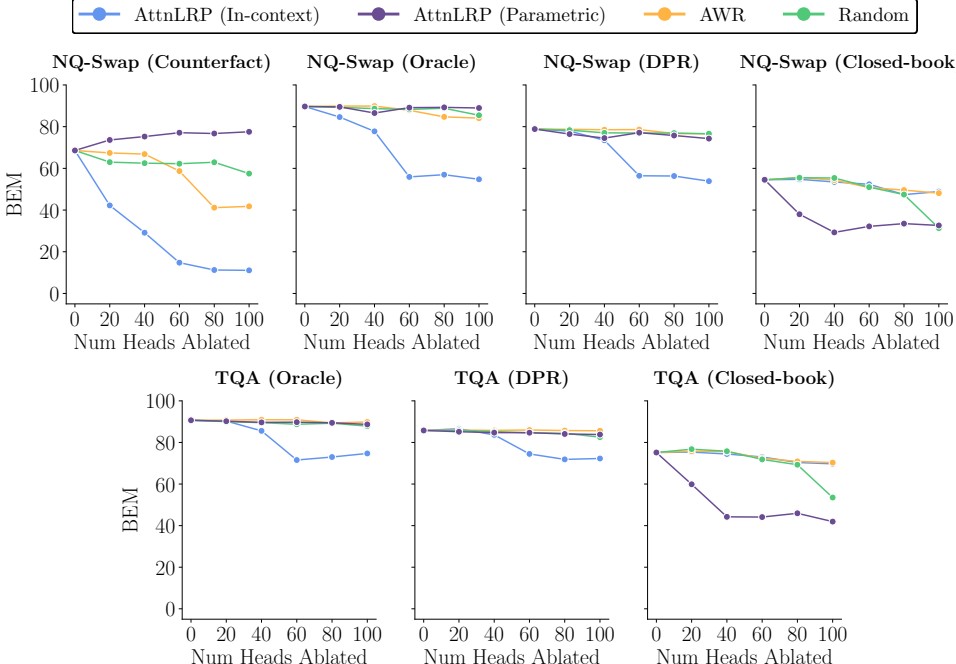

Figure 18: BEM score analysis for Gemma-2-9B-it when either in-context or parametric heads are ablated on NQ-Swap and TQA under various configurations. For DPR, we use instances where K-Precision [3] is equal to 1 in the non-ablated run.

