# OpenReview forum: "The Atlas of In-Context Learning: How Attention Heads Shape In-Context Retrieval Augmentation"
_NeurIPS.cc/2025/Conference — NeurIPS 2025 poster_

### Official Review · Reviewer_ZnzS · 2025-06-03

**Clarity:** 3
**Significance:** 3
**Originality:** 3
**Rating:** 5
**Confidence:** 4

**Summary:**

This paper provides a mechanistic analysis of how large language models retrieve information in context. The authors employ an attribution-based methodology to identify distinct sets of attention heads that serve different functions: parametric heads that extract knowledge from model parameters, task heads that process task-specific information, and retrieval heads that extract information from the given context. They validate the functionality of these head categories through ablation studies that demonstrate their causal roles in the model's behavior.

**Questions:**

**Q1.** Regarding the statement "In addition, we use counterfactual contexts as they contain information that is usually not seen during pre-training and fine-tuning stages, thus forcing models to rely on the text to answer the question correctly" (p.4, l.149-150), could you clarify where the counterfactual contexts are sourced from?

**Q2.** You mention that "this leads to a slight performance increase on NQ-Swap with counterfactual contexts, which suggest that the ablation forces the model to rely more on the given context instead of its own parametric knowledge" (p.6, l.209-210). Depending on your counterfactual data selection method, do you verify that the context and parametric answers differ? This result suggests that removing parametric heads eliminates knowledge conflict and improves performance - it would be valuable to discuss this knowledge conflict aspect more explicitly.

**Q3.** What happens qualitatively when you ablate in-context heads versus parametric heads? When ablating in-context heads, does the model default to parametric knowledge (especially during knowledge conflicts)? What occurs when ablating parametric heads in the CB example? What happens if you ablate both types?

**Q4.** In Figure 1, what exactly do the bar plots represent? Are they showing average task/retrieval relevance for all heads in that layer, or the maximum? Is the marker size representing each head's relevance, thresholded by some minimum value?

**Q5.** Why would task heads be marked as context heads or have particularly high context scores? According to Equation 5, these heads should be relevant for both CB and OB, so they should have values around zero. Do you have intuitions for explaining this pattern?

**Q6.** Since $H_{task}$ and $H_{ret}$ are not mutually exclusive, did you examine whether some heads have high values in both categories?

**Q7.** For the experiments in Section 6.2:
- Are you using the chat template for the instruct models? Figure 4 appears to use a Q: ... A: ... format, which seems uncommon and potentially out-of-distribution for chat models.
- Could you clarify the experimental setup for modifying attention weights? The "Attn Weight" label is only relevant for $\cal{H}_{ret}$ right?
- Did you evaluate model performance without any patching? I expect it to be similar to random, but might still be interesting.

**Q8.** You state that "Oracle context is often not available; therefore we also use Dense Passage Retriever (DPR)" (p.4, l.151), and Figure 2 only reports Oracle results. Does Figure 2 only show examples where true Oracle context exists, or does it refer to DPR? The appendix displays DPR separately, which creates some confusion.

**Q9.** While the results for $H_{param}$ are good they are clearly worse than the other two sets. Can you provide any intuitions of why you see poorer performance here? My hypothesis would be that different heads deal with different domains of parametric knowledge, whereas this may be less the case for contextual knowledge. This would mean that, in order to cover many domains, you would just need more parametric heads.

**Expectation Management:** I think this is a great paper with just some minimal issues. If you can resolve W1/Q1 & W2 (without reproducing the other paper) as well as clarify the confusion around W7 & W8, I'll happily raise the score to 5. If you manage to reproduce/explicitely quantitatively compare to https://aclanthology.org/2024.findings-acl.70.pdf, and include discussions about Q2/W13 as well as all minor issues, I will consider a score of 6. Even without addressing Q2/W13, I'd strongly recommend implementing the presentation improvements W4, W5, W6, W9, W10, and W11 for CR as they would enhance the paper's clarity.

**Ethical Concerns:**

["NO or VERY MINOR ethics concerns only"]

**Final Justification:**

The authors addressed all my concerns and questions to my satisfaction, which is why I increased my score as mentioned under 'Expectation Management'. I refrain from giving a score of 6 because, while I believe this paper is very solid and has a significant impact, I don't think it warrants the term 'groundbreaking impact'. Secondly, the authors were unable to provide a direct comparison to the work of Jin et al. [1]. Nevertheless, I recommend that this paper is accepted, as I believe it would make a valuable contribution to NeurIPS.


[1] Jin, Zhuoran, et al. "Cutting off the head ends the conflict: A mechanism for interpreting and mitigating knowledge conflicts in language models." Findings of ACL (2024).

**Limitations:**

yes

**Quality:**

4

**Strengths And Weaknesses:**

## Strengths

**S1.** Great idea with the synthetic document dataset approach.

**S2.** Good visual presentation with effective use of color and high-quality figures that effectively communicate the results (aside from font size issues).

**S3.** The results in Section 6.2 are impressive and demonstrate strong empirical validation.

**S4.** The paper addresses most questions that arise during reading, providing comprehensive coverage.

## Weaknesses

### Major

**W1.** The method for generating counterfactual data needs clearer explanation.

**W2.** The relationship to the prior work https://aclanthology.org/2024.findings-acl.70.pdf requires better contextualization. While you cite this work, you dismiss it as only looking at single word settings. However, the parallels are substantial - they also compute two sets of heads. Your statement that "Our method identifies two groups of heads based on their functions: parametric heads that encode relational knowledge and in-context heads responsible for processing information in the prompt" (p.1, l.33-34) needs better positioning relative to existing work. Consider comparing methods directly. Demonstrating that you find similar heads or that your head sets perform better would be optimal (but not necessarily required, see *Expectation Management* under *Questions*).

### Minor

**W3.** While you discard inhibitory effects, there likely exists interplay where parametric heads inhibit context answers and vice versa. Did you explore this? This might be worth putting into the limitations.

**W4.** Figure fonts are generally too small across multiple figures (Figures 1, 2, axis labels). Consider increasing font sizes where space permits.

**W5.** Figure 1 should be exported as a vectorized image for better quality.

**W6.** Minor notation issue: in my opinion "for a sequence S" (p.4, l.127) should be "sequence of length S".

**W7.** The statement "retrieval heads and task heads consistently rank among the highest scoring in-context heads, emphasizing the critical role of in-context heads for retrieval augmented generation" (p.7, l.262-266) appears overstated. Figure 2 shows that ablating only 20 heads deteriorates performance, while most retrieval heads rank between 100 and 400.

**W8.** The hyperparameter K mentioned as "will be determined in the next section" (p.7, l.243) is never formally determined - you simply state that you use top 40 heads (l.260). Explicit definition and justification for K selection would improve clarity. The appendix brings clarity, but then you must refer to that. The same holds for Table 1, where it is completely unclear how many heads you compare. From reading the main text (without looking at the appendix) the conclusion would be that you compare 100 $H_{param}$ heads to 40 of the other heads. From reading the appendix, I assume you're using 40 $H_{param}$ heads as 50 heads already performs better for Llama. You should be more specific about this.

**W9.** Section 6.2 should mention the existence of the FV scaling factor $\alpha$ in the main text (footnote).

**W10.** Experiment 7 is difficult to understand without consulting Appendix D. Consider expanding the explanation, focusing on your proposed metrics rather than the logit lens function. I claim that most interested readers will be familiar with logit lens. If you get another page for camera ready, I'd recommend spending extra lines explaining this experiment. Further, Figure 5b doesn't visualize the weights, which are important here.

**W11.** Table 2 presentation could be improved. In my understanding, ROC AUC measures parametric vs contextual knowledge, while other metrics measure localization performance. Consider separating these with vertical lines or hierarchical headers for clarity.
Given the density of this section, placing these metrics adjacent to each other initially creates confusion as readers may attempt to compare them inappropriately.

**W13.** You are missing an opportunity to report additional metrics on resolving knowledge conflicts in the counterfactual setting. While you demonstrate that ablating context heads destroys recall, you don't discuss what happens to the parametric answer. If you could show that your metric enables outperforming other knowledge conflict interpretation papers (those cited on l.59), this would significantly strengthen your contribution.

---

> ### Author Rebuttal · Authors · 2025-07-31
>
> We are truly grateful to the reviewer for their exceptional feedback. This is genuinely the most insightful review we have ever received in any conference setting we participated in so far. Your comments have substantially improved our presentation, and we deeply appreciate the care you invested in engaging with our work.
>
> **[Counterfactual data W1/Q1]**
> Thank you for the opportunity to clarify. We use NQ-Swap, a variant of Natural Questions based on the MRQA 2019 task. The counterfactual data is generated through corpus substitution, where QA instances with named entity answers are identified. We then replace all occurrences of the named entities in the context by other entities sampled from the NQ dataset which have the same type as the original. This is intentional to elicit knowledge conflicts and can be used to evaluate whether LLMs rely more on provided context or on memorized knowledge learned during pre-training (since the substitute entities usually do not appear in the same context during pre-training). We will revise the manuscript to explain this more clearly.
>
> **[Relationship to prior work W2]** We believe the work of Jin et al. [3] is a well-executed contribution to the field. We appreciate their effort in identifying parametric and in-context heads and in exploring head-level ablations to mitigate knowledge conflicts. However, we note that their study is only tangentially related to ours: specifically to our initial experiment in Section 5, where we identify in-context and parametric heads. In our case, this experiment serves primarily as a first step for the more substantive contributions in the subsequent sections. There are several key differences:
>
> - First, Jin et al. focus on narrow-domain tasks with single-token answers, which diverges from realistic QA use cases. In contrast, we conduct our analysis on open-domain QA datasets (NQ-Swap, TriviaQA), as well as a curated human biography dataset that better captures the complexity of real-world settings.
> - While we would have welcomed a direct comparison with Jin et al., this is unfortunately not possible. Their GitHub repository is empty, and no code is available. More importantly, their method—based on path patching [5]—is computationally impractical for QA tasks, requiring 2048 forward passes per sample (1024 heads × 2 passes), which would take 21 days for a single test set run over NQ-Swap on an RTX 4090.
> As discussed in Section 3 (L82–94), we surveyed various attribution methods and found limitations in scalability or faithfulness. We selected AttnLRP [1] for its competitive attribution quality and significantly lower computational cost. Although Jin et al. briefly explored gradient-based methods (§4.3), they dismissed them due to lack of faithfulness. This criticism applies to standard gradient-based approaches, which are indeed known to suffer from high noise, a limitation well-documented in the literature[1,2]. However, AttnLRP improves on gradient-based methods, hence we selected it.
> - For the first time, we show that in-context heads are considerably more complex and can specialize to task or retrieval heads which provides a finer-grained categorization than is available in literature. We also empirically demonstrate their causal roles in shaping the model’s representation (via intensional frames) along with parametric heads.
> Our method allows us to extract steerable function vectors in a zero-shot manner for complex tasks, as described in (Section 6.2).
> - We also introduce a novel post-hoc procedure to trace both contextual and parametric knowledge usage in LLMs, thus enabling the identification of potential hallucinations by providing citations (Section 7), especially since this line of work has become increasingly important [4].
>
> **[Inhibitory effects W3]** We indeed identify inhibitory heads that appear to suppress the influence of counterfactual inputs, reinforcing the model’s parametric answer. For the rebuttal, we extended our analysis to task heads and found similar inhibitory behavior. Notably, patching such heads fails to induce the intensional frame, likely because they destabilize it. Boosting their activation yielded inconsistent effects: sometimes suppressing the frame, other times increasing verbosity. Overall, the patterns are difficult to interpret without a more systematic analysis and we leave it for future work. We will also put this in the limitation section.
>
> **[Fig. W4+W5, notation issue W6, scaling factor W9, section 7 W10]** We sincerely thank you for your detailed feedback and will gladly incorporate these improvements into the manuscript.
>
> **[Ranking of heads W7]** What we intended to convey is the reverse relationship: as shown in Fig. 3, the highest-scoring in-context heads are primarily composed of retrieval and task heads. We did not mean to imply that all retrieval heads rank among the top in-context heads. We will revise this to more accurately reflect the intended meaning.
>
> **[K-Hyperparams W8, Table 2 W11]**
> To save space, we moved non-essential details to the App. and briefly noted this in L279. The table is also compact for space reasons. We will expand on this in the camera-ready version and we appreciate the helpful suggestion.
>
> **[Knowledge Conflict Q2 & W13]**
> Thank you for this fruitful suggestion. We computed the accuracy of LLama 3.1 on the NQ-Swap with counterfactual contexts. Computing the recall score on the original gold answer instead, the model is clearly distracted by the wrong contexts achieving only 9% accuracy. Removing the top 60 in-context heads reduces the reliance on the wrong context pushing the score to 35% which is similar to the CB accuracy of the model. Conversely, computing the recall score for counterfactual answers instead, the model achieves 68% accuracy without ablation. Ablating the top 100 parametric heads reduces the reliance on internal knowledge and the model is more likely to accept the counterfactual context increasing the score to 76%. In cases where the model still fails, it often asserts that the answer is not present in the context indicating a complete failure to locate it, unrelated to parametric memory.
>
> **[Qualitative analysis for ablation Q3]**
> - The model tends to revert to parametric knowledge when in-context heads are ablated (/w counterfactual contexts), while sometimes returning incorrect but semantically related answers or “The text does not mention this information.”
> - When parametric heads are ablated in CB, the model tends to return false but semantically plausible answers or mention “I do not have information on …”.
> - When both heads are ablated, models seem unable to read from the contexts (/w counterfactual) and hallucinate semantically plausible answers (counterfactual and CB).
> We will add some qualitative examples.
>
> **[Bar Plots & Marker Size Q4]**
> We agree this aspect could be clearer and we will revise it accordingly. The bar plot (Section 6.1 L246–248) shows layer-wise relevance distributions by summing, per layer, the relevance each head assigns to the question or answer object tokens. This reveals whether a layer is more task- or retrieval-focused. Top 40 task and retrieval heads are highlighted with larger markers, while marker color indicates in-context vs. parametric behavior (based on the top 100 heads in each category).
>
> **[Task heads in OB and CB Q5]**
> As shown in Fig. 3 & 6, while some task heads show strong in-context behavior, many fall in the moderate range. Notably, Gemma 2 even exhibits parametric task heads, indicating that task heads are not exclusively in-context. We will clarify this in the revised version. As for why task heads receive more relevance in OB settings: one explanation is that they must handle more contextualization, thereby attracting more relevance. In contrast, in CB settings, parametric heads dominate as the input is more self-contained and absorb more relevance.
>
> **[Exclusivity of heads Q6]**
> Task and retrieval heads are largely mutually exclusive. For example, in LLaMA, the intersection between the top 40 task heads and the top 40 retrieval heads yields only a single shared head. However, we believe there may be a small misunderstanding: in App. (L803–811), we state that in-context heads and task heads tend to show considerable overlap (not retrieval heads). We will clarify this more explicitly.
>
> **[Chat template, attention modification, performance w/o patching Q7]**
> We did not use chat-specific templates, but followed [6] to exclude possible system prompt interference.
> Exactly, only for the evaluation of retrieval heads did we modify the attention weights. This is explained in App. L856-867.
> We evaluated the performance without any patching in our preliminary experiments and it indeed yielded scores slightly below random.
>
> **[Fig. 2 Q8]** Indeed, it refers to true oracle and not DPR. We will add this to the main text.
>
> **[Performance of H_param Q9]** Your hypothesis appears correct. We patched parametric FVs from cloze-style statements into related question prompts (e.g., “Mike Tyson was born on” → “Q: What is the birth date of Albert Einstein?”). On Llama, we see an average increase of ~7% with 100 heads, which confirms that more parametric heads are needed to cover many domains.
>
> Ref:
>
> [1]  Achtibat, Reduan, et al. "AttnLRP: Attention-Aware Layer-Wise Relevance Propagation for Transformers." ICML, 2024.
>
> [2] Smilkov, Daniel, et al. "Smoothgrad: removing noise by adding noise." arXiv preprint  (2017).
>
> [3] Jin, Zhuoran, et al. "Cutting off the head ends the conflict: A mechanism for interpreting and mitigating knowledge conflicts in language models." ACL Findings (2024).
>
> [4] Qi, Jirui, et al. "Model internals-based answer attribution for trustworthy retrieval-augmented generation." EMNLP (2024).
>
> [5] Goldowsky-Dill, Nicholas, et al. "Localizing model behavior with path patching." arXiv preprint (2023).
>
> [6] Ram, Ori, et al. "In-context retrieval-augmented language models." ACL (2023)

---

> > ### Comment · Reviewer_ZnzS · 2025-08-01
> >
> > Thanks for your detailed answers. I’m mostly happy with them, except for the following comments:
> >
> > **W2:** While I agree with all of these points, the work is similar enough that it deserves a more detailed reference than just “Some works have studied mechanisms behind knowledge preference in RALMs.” That sentence fits well for the other papers cited, but I think you should introduce that paper specifically and summarize the key differences, which definitely exist, as you have shown.
> >
> > **W13:** Cool! Even if these results are not exceptional, I think they are worth a brief discussion in the appendix.
> >
> > **Q3:** Same here—it would be nice to include this in the appendix.
> >
> > **Q7:** I find the choice of chat template a bit unusual; I’d encourage you to clarify this choice in the text (or a footnote).
> >
> > My main issue remains W2. I’ll gladly increase to 5 once you’ve addressed it appropriately.

---

> > > ### Author Response · Authors · 2025-08-03
> > >
> > > Thank you for your fast response. We apologize for any confusion caused by character limits. To clarify, we of course will incorporate all the points raised in our rebuttal into the revised manuscript. To be more specific, we will:
> > >
> > > **W2**: Extend the related work section, to contextualize and summarize the key differences of our work with Jin et al., [1]. Furthermore, we will emphasize that earlier studies were largely limited to toy settings, and we contribute by extending this line of research with a more comprehensive and fine-grained evaluation of how parametric and in-context heads that have been previously identified may specialize – by looking at prompt components and applying attribution score filtering, function vector extraction, and targeted attention weight modifications.
> > >
> > > **W13**: Incorporate discussion of knowledge conflict results in the Appendix. We would like to point out that we use a multi-token instead of a single-token answer setting as in prior works. While our results are not directly comparable to prior works due to this choice, we believe it strengthens our contribution since it is closer to real-life scenarios.
> > >
> > > **Q3**: Incorporate the additional qualitative ablation analysis in the Appendix.
> > >
> > > **Q7**: Clarify the choice of chat template in the main text.
> > >
> > > We have addressed all the concerns in the rebuttal (especially regarding critical points W1/Q1, W2, W7, W8, Q2/W13 as well as all minor issues wrt. your expectations), which as far as we understand, satisfies your inquiries. Unfortunately, direct comparison with Jin et al., [1] would be impossible due to the lack of available code, which is beyond our control.
> > >
> > > Given these considerations and the scope of our contributions, we kindly ask whether a score of 6 is still being considered on your end. We look forward to further discussion and would be happy to address any additional questions you may have.
> > >
> > > References:
> > >
> > > [1] Jin, Zhuoran, et al. "Cutting off the head ends the conflict: A mechanism for interpreting and mitigating knowledge conflicts in language models." Findings of ACL (2024).

---

> ### Comment · Reviewer_ZnzS · 2025-08-04
>
> Thanks — this addresses all my concerns. I believe this is now a solid paper that should be accepted. I will update my score to 5 and defend this stance during the reviewer discussion. I've also increased the clarity score, considering you correctly implement all the discussed improvements. I won't consider a score of 6 because of the missing direct comparison, even though I agree that missing code is a problem. I also believe that the paper's impact doesn't classify as  'groundbreaking' (a score of 6 is described as 'Technically flawless paper with groundbreaking impact on one or more areas of AI [...]').

---

> > ### Author Response · Authors · 2025-08-04
> >
> > Thank you for raising your scores and acknowledging our rebuttal and the missing code problem. We sincerely appreciate your thoughtful review and valuable suggestions.

---

### Official Review · Reviewer_FEvZ · 2025-07-03

**Clarity:** 2
**Significance:** 1
**Originality:** 2
**Rating:** 3
**Confidence:** 3

**Summary:**

This paper investigates the mechanism behind in-context retrieval augmentation by identifying attention heads specialized for this job in large language models. Using an attribution-based method, AttnLRP, the authors categorize attention heads into two types: in-context heads that process contextual information from prompts, and parametric heads that store relational knowledge acquired during pre-training. The paper shows that in-context heads further specialize into task heads (for instruction following) and retrieval heads (for pure copy-paste from context). Through controlled experiments on biography datasets and QA tasks, the authors demonstrate that these heads can be compressed into function vectors or have their attention weights modified to causally influence answer generation. The work concludes by presenting a linear probe that can distinguish between parametric and contextual predictions with high accuracy and claim to enable more transparent analysis on context based generation.

**Questions:**

Q1: Please correct me if my understanding is wrong. I’m wondering if it is surprising that attention tracing for the answer token does localize the token which probably is similar in the context in Table 2. Am I mis-understanding this experiment?

**Ethical Concerns:**

["NO or VERY MINOR ethics concerns only"]

**Final Justification:**

I confirm my final score of 3.

While this paper shows some promising results, the practicality of the results compared to simple baselines such as prompting doesn't seem to be clearly demonstrated.

I believe that the assigned score is adequate.

**Limitations:**

Yes.

**Paper Formatting Concerns:**

None.

**Quality:**

3

**Strengths And Weaknesses:**

S1: This paper addresses an interesting topic of interpreting what attention heads of a transformer perform in different roles. The experimental design reveals the existence of attention heads delivering parametric and in-context knowledge, providing valuable insights into the mechanistic understanding of retrieval-augmented generation.

S2: The paper presents a relatively thorough investigation with many well-designed control experiments, including ablation studies, function vector extraction, and cross-dataset generalization tests that strengthen the empirical findings.

S3: The attribution-based methodology using AttnLRP provides a principled approach to head identification that goes beyond simple attention weight analysis.

---

W1: The experiments are sound, but it seems like there aren't many actionable items that practitioners could directly use. This would be enhanced if the whole method could be efficiently packaged to apply directly to any transformer-based LM, or if performance gains on some realistic benchmarks could be achieved through the insights gained.

W2: It seems like most experiments still rely on single-token answers, and it is unclear how these findings could help with any interpretability of answers with long chain-of-thought reasoning, which represents the vast majority of LLM use cases in practice.

W3: I think interpretability studies can benefit from showing real-life impact, as pointed out in recent discussions (https://icml.cc/virtual/2025/workshop/39962, this is just showing that this discussion is significant enough to grow into a workshop). However, I cannot directly see a direct impact of this study beyond advancing our theoretical understanding of attention mechanisms. To this end, can this interpretability method be at least compared to simple prompting: Literally asking the model “Why did you answer ###? Is there a reason from the context?”

W4: It seems like some important citations on mechanistic interpretability are missing: https://arxiv.org/pdf/2310.10348, https://arxiv.org/pdf/2407.00886

W5: Correct me if I am wrong, the method seems like a direct application of AttnLRP, or are there improvements delivered?

---

> ### Author Rebuttal · Authors · 2025-07-30
>
> We sincerely thank the reviewer for their encouraging feedback and we are especially grateful for the recognition of our contributions to the mechanistic understanding of retrieval-augmented generation (S1), as well as the acknowledgment of our experimental rigor through comprehensive and well-designed experiments (S2). We also appreciate the reviewer’s appreciation of our attribution-based methodology as a principled and meaningful advancement over prior studies (S3). In the following, we respectfully address the reviewer’s concerns and suggestions in the responses below. We hope our clarifications and revisions further strengthen the contributions and clarity of the work.
>
> **[Sound, but no actionable items (W1) and benefit for real-life impact (W3)]**: We thank the reviewer again for their thoughtful feedback and for acknowledging the soundness of our experimental design. We fully agree that interpretability research should strive to connect with real-world applications that benefit practitioners. Toward this goal, we have aimed to present not only a theoretical advancement in understanding attention head specialization in LLMs, but also practical implications that enhance model transparency and safety.
>
> Specifically, we **added additional experiments on detecting knowledge conflicts** for Section 7: scenarios in which LLMs must choose between conflicting contextual (external) information and their internal parametric memory [1]. As shown in the paragraph below, our method allows us to guide the model to prefer either contextual knowledge or its memorized answer, depending on the desired use case. As such, a practitioner could provide a high quality knowledge base for a RAG system e.g. legal texts and ground the model on this external information by preventing the LLM from using outdated internal knowledge. This demonstrates how our method can be used to detect and mitigate knowledge conflicts, an important step toward improving LLM reliability.
>
> In addition, we respectfully draw attention to Section 7, where we leverage our findings to build faithful post-hoc citation mechanisms. We introduce a novel attribution-based procedure for tracing both contextual and parametric knowledge usage within LLMs, enabling the creation of **faithful citations** and hence introducing a control mechanism **against hallucinations**, which is a major problem, particularly given its growing importance in the NLP communities [2, 3].
>
> **[Knowledge conflict results]**: We computed the accuracy of LLama 3.1-8B-Instruct on the NQ-Swap with counterfactual contexts. Computing the recall score on the gold answer instead, the model is clearly distracted by the wrong contexts achieving only 9% accuracy. Removing the top 60 in-context heads reduces the reliance on the wrong context pushing the score to 35% which is similar to the closed-book accuracy of the model. Conversely, computing the recall score for counterfactual answers i.e. reliance on the counterfactual inputs, the model achieves 68% accuracy without ablation. Ablating the top 100 parametric heads reduces the reliance on internal knowledge and the model is more likely to accept the counterfactual context increasing the score to 76%.
>
> **[Direct applications to Transformer LMs (W1)]**: Yes, our method is model-agnostic and can be readily applied to any pretrained LLM without requiring fine-tuning or architectural modifications. Furthermore, it is straightforward to package this approach into a reusable library. We will open source our implementation to facilitate this.
>
> **[Comparability to simple prompting (W3)]**: As noted in L60–61, RAG LLMs cannot guarantee faithful attribution to contextual passages [2], highlighting the need for alternative methods such as ours for post-hoc knowledge tracing. This perspective also aligns with recent works, e.g. by Qi et al. [3], which emphasizes the value of actionable interpretability as it provides actionable insights for both users and developers of LLM systems.
>
> Regarding prompting-based attribution, we also note that LLMs **do not reliably** reflect the true origin of their outputs. For instance, they may exhibit **confirmation bias** when asked leading questions such as *“Is there a reason in the context...”*. Even when the external context conflicts with internal memory, models often confirm answers aligned with parametric knowledge, as shown in [4]. [5] also demonstrates that LLMs strongly prefer generated contexts regardless of correctness, which can still suffer from hallucinations . These findings make the reliability of prompting heuristics questionable and support our argument that interpretability methods are essential for faithful knowledge attribution and conflict resolution.
>
>
> **[Single-token answers (W2)]**: We use multi-token answers for the vast majority of our experiments and highlight how this differs from previous works in the related work section. We will make this clearer in the revised version of our manuscript.
>
> **[Missing citations (W4)]**: Thank you for the references. We will add this to the revised version of our work.
>
> **[Direct applications of AttnLRP (W5)]**: While we opt to use AttnLRP due to our need for a method that is both computationally efficient and faithful in attributing model behavior (L82-94, L105-L103), our framework is general and can be applied to other feature attribution methods.
>
> **[Is it surprising that attention tracing for answer tokens does well for localization? (Q1)]**: Thank you for the thoughtful question. At a first glance, it may indeed seem normal that an answer token attends to contextually similar tokens. However, our work goes beyond this intuition in several important ways.
>
> First, we do not simply visualize attention weights. We identify **which specific** attention heads are responsible for tracing the origin of an answer, namely the retrieval heads. Using attribution analysis, we show that only a **small subset** of heads (4% of total heads) meaningfully contribute to source tracing, while others are largely inactive or fulfill orthogonal roles. This is a nontrivial finding: constructing accurate relevance heatmaps requires knowing **which** attention heads to include and how to **weight** their contributions correctly, as described in Section 7 and Appendix D. Without this guidance via attribution, one would **not** obtain the sharply focused heatmaps we present, but instead **noise**. As discussed in Section 3, this selective identification is not achievable through raw attention maps alone. Attribution methods are therefore essential: they reduce the search space and expose the functional roles of heads, rather than assuming all attention is equally explanatory.
>
> Second, we recognize the broader interpretability debate, particularly the concern that “Attention is not explanation” [6]. Our approach addresses this challenge directly: rather than assuming attention weights are inherently meaningful, we **validate empirically** that certain attention heads produce faithful and reproducible attribution signals, while others do not.
>
> Crucially, our method requires no fine-tuning, supervision, or model modifications, distinguishing it from prior approaches to source attribution that rely on explicitly training models for this purpose [7]. Our **zero-shot, post-hoc** framework is therefore broadly applicable and practical for analyzing existing models without additional overhead for practitioners.
>
> We thank the reviewer again sincerely for the time spent engaging with our work and we hope our responses have clarified the concerns raised.
>
> References:
>
> [1] Xu, Rongwu, et al. "Knowledge conflicts for LLMs: A survey." arXiv preprint arXiv:2403.08319 (2024).
>
> [2] Gao, Tianyu, et al. "Enabling large language models to generate text with citations." EMNLP (2023).
>
> [3] Qi, Jirui, et al. "Model internals-based answer attribution for trustworthy retrieval-augmented generation." EMNLP (2024).
>
> [4] Xie, Jian, et al. "Adaptive chameleon or stubborn sloth: Revealing the behavior of large language models in knowledge conflicts." The Twelfth International Conference on Learning Representations. 2023.
>
> [5] Tan, Hexiang, et al. "Blinded by generated contexts: How language models merge generated and retrieved contexts when knowledge conflicts?." arXiv preprint arXiv:2401.11911 (2024).
>
> [6] Bibal, Adrien, et al. "Is attention explanation? an introduction to the debate." Proceedings of the 60th Annual Meeting of the Association for Computational Linguistics (Volume 1: Long Papers). 2022.
>
> [7] Sun, Zhiqing, et al. "Recitation-augmented language models." arXiv preprint arXiv:2210.01296 (2022).

---

> > ### Author Response · Authors · 2025-08-05
> >
> > Dear Reviewer FEvZ,
> >
> > we have replied to your questions. As the deadline is approaching, could you please respond at your earliest?
> >
> > Looking forward to further discussion and also happy to address any additional questions you may have.

---

> > ### Comment · Reviewer_FEvZ · 2025-08-05
> >
> > W1, W3: I agree that the method works. However I believe the debate should be whether the method is necessary or overcomplicated compared to other means to achieve the goals. To be concrete, I am wondering if the control on what information to use or where the answer is sourced from is truly impossible via simply prompting the model. If it could be achieved, why would one use a more complex method? I'm not trying to be unfair for mechanistic interpretability methods, I am in fact trying to be fair to all methods, including "simply ask the model", which should be a good baseline even if it doesn't seem technical.
> >
> > And as for the authors comment on prompting methods: I agree that prompting is not perfect either. However these papers the authors cited are indeed pointing out that it could be enhanced, and not that prompting never works and cannot be made better. I agree that they show "confirmation bias", but how can I know if the model internal based method doesn't also have some form of bias? With the current experiments, I cannot get an apples to apples comparison.
> >
> >
> > Q1: I am not questioning how different the method is from just attention maps. I am asking if this result is unexpected or surprising? It seems very natural to me that tracing the answer token does indeed lead to the token in the context. I appreciate the contribution to the debate "Is Attention attribution" (with the caveat that most researchers already seem to no that its in general not equivalent), but Table 2 doesn't seem to significantly alter preexisting beliefs.

---

> > > ### Author Response · Authors · 2025-08-06
> > >
> > > Thank you for your prompt response! We highly appreciate your stance regarding the soundness of our method.
> > >
> > > **[Is the method necessary or overcomplicated to achieve the goals compared to prompting? (W1, W3)]**
> > >
> > > While prompting the model indeed looks simpler compared to our method, we argue that due to its nature as a *black-box* intervention (which will still persist even after possible improvements), it cannot achieve our goal to illuminate the mechanism of in-context retrieval augmentation. It may give insights into *what* the model does, but not *how* this is achieved in terms of its component. Prompts are also extremely sensitive to subtle changes [1] which in turn affects which information to use, hence potentially producing brittle or even false explanations orthogonal to our aim.
> > >
> > > The aforementioned points above necessitate a glass box interpretability approach. In this regard, our method allows inspection of models’ internal computations e.g., attention patterns, head activations, relevance flow and can *trace* how specific outputs emerge from specific components. This approach is fundamentally more *objective* and *reliable*. It does not rely on a model's self-report (which can possibly be misleading or hallucinated [2]), since it gives direct access to the underlying mechanisms.
> > >
> > > Regarding bias, we agree that it is a fair and important question. While we cannot be 100% certain, as is common with most methods in this field, we have taken care to verify our method experimentally through ablations, performance evaluations, and other quantitative checks which gives us strong confidence that we have captured a core part of the mechanism. We believe our findings are robust, rigorously validated, and provide meaningful insight into the model’s behaviour and thus making them worth sharing with the community.
> > >
> > > With that being said, we fully acknowledge that the model may also implement redundant or parallel circuits that we haven’t uncovered yet. We will add this point to the limitation section. We also agree that it would be useful to include prompts as a baseline and we’re happy to explore that in the future. However, we would like to point out that LLMs often hallucinate or produce misleading justifications, which is well-documented and limits the reliability of prompting-based control to retrieve answer sources [3,4, *inter alia*].
> > >
> > > **[Unexpected or surprising results wrt. attention maps Q1]**
> > >
> > > Thank you for the comment. Our experiments in Section 7 and the Appendix are intended to confirm the reliability of our method. We believe that our results provide an interesting perspective: The identified retrieval heads are surprisingly performant in roughly approximating the explanatory power of AttnLRP wrt. models’ ability to copy text verbatim. This means while in general attention is not equivalent to attribution as you mentioned, our results demonstrate the possibility to turn them into viable explanations which we have shown to be useful on hallucination detection and source attribution. From a practical perspective, this also reduces the computational cost since only attention maps are required.
> > >
> > > From our side, we believe we have successfully addressed the concerns raised in W1, W2, W4, and W5. As for W3/Q1, we believe this rather reflects broader limitations common to interpretability as a whole and not weaknesses of our method.
> > >
> > > We would be happy to address any additional questions you may have. If our comment addresses your concerns, we would greatly appreciate your consideration in updating the score.
> > >
> > > References:
> > >
> > > [1] Sclar, Melanie, et al. ”Quantifying Language Models' Sensitivity to Spurious Features in Prompt Design or: How I learned to start worrying about prompt formatting.” ICLR (2024)
> > >
> > > [2] Yao, Jia-Yu, et al. “LLM Lies: Hallucinations are not Bugs, but Features as Adversarial Examples.” arXiv preprint arXiv:2310.01469 (2024).
> > >
> > > [3] Liu, Nelson, et al. “Evaluating Verifiability in Generative Search Engines.” Findings of EMNLP (2023).
> > >
> > > [4] Gao, Tianyu, et al. "Enabling large language models to generate text with citations." EMNLP (2023).

---

### Official Review · Reviewer_n2tY · 2025-07-03

**Clarity:** 3
**Significance:** 3
**Originality:** 3
**Rating:** 4
**Confidence:** 3

**Summary:**

It proposes an attribution-based method to identify attention heads that are crucial during in-context retrieval augmentation, showing that these heads operate distinctly on the prompt. This method can categorize attention heads into "in-context" and "parametric" types.
It analyzes how in-context heads specialize in interpreting instructions and retrieving information, mapping their locations across different model layers.  Furthermore, it demonstrates how both in-context and parametric heads influence the answer generation process by compacting them into function vectors or modifying their attention weights.
The research presents preliminary findings on enabling causal tracing of source information for retrieval-augmented language models, which indicates promising directions for improving the interpretability of Retrieval-Augmented Generation (RAG) systems.

**Questions:**

1. Methodological Specificity and Generalizability (of the attribution method):
The paper proposes an "attribution-based method" to identify specialized attention heads. My question would be: What are the precise mathematical definitions and computational procedures of this method? How robust and generalizable is this approach across different Transformer architectures or even different languages beyond English?

2. Boundaries and Interactions of Attention Head Classification:
The paper categorizes attention heads into "in-context" and "parametric." Are these classifications absolute, or is there a spectrum or potential for "hybrid" heads? Furthermore, how do these identified attention heads interact and cooperate with other core components of the model, such as the Feed-Forward Networks (MLPs), which are also known to store knowledge?

3. Universality of Insights and Limitations in Application:
The insights presented are primarily derived from in-context learning in question-answering tasks. To what extent do these findings regarding attention head function and influence universally apply to other types of in-context learning tasks (e.g., summarization, translation, code generation), or remain valid in more complex multi-modal and multi-task learning scenarios? What are the potential limitations of these findings in practical applications, such as model fine-tuning or safety auditing?

**Ethical Concerns:**

["NO or VERY MINOR ethics concerns only"]

**Final Justification:**

Accept ---
The authors have provided a clear and thorough rebuttal, directly addressing my concerns and those of the other reviewers. They have committed to adding clarifications, improving presentation, expanding the related work section, and including additional results and analyses that will strengthen the paper. Given the novelty of their mechanistic analysis on in-context retrieval augmentation, the solid empirical evidence, and the promising implications for building more transparent and trustworthy RAG systems, I am satisfied with their responses and recommend acceptance.

**Limitations:**

1. Limited Scope of Component Interaction Analysis: The study primarily focuses on the role of attention heads. It acknowledges that the full picture of how attention heads interact with other crucial components of LLMs, such as MLP (Multi-Layer Perceptron) modules (e.g., knowledge neurons), to induce specific functions remains an open question and is left for future investigation.

2. Potential for Misuse of Insights: The paper highlights a broader ethical concern regarding the potential for the identified specialized heads to be misused, for instance, to induce or amplify malicious model behaviors. This is a general ethical limitation tied to deep interpretability, rather than a methodological flaw.

3. Untested in Diverse In-Context Learning Scenarios: While the paper demonstrates the mechanism in retrieval-augmented question answering, its findings are not explicitly tested across a wider variety of in-context learning tasks (e.g., in-context summarization, translation, or more complex reasoning tasks) where retrieval might also play a role, or in multi-modal contexts.

**Quality:**

3

**Strengths And Weaknesses:**

Strengths:
1. Pioneering Attention Head Classification: The paper introduces a novel attribution-based method that effectively identifies and clearly categorizes attention heads into "in-context" and "parametric" types. This is a crucial breakthrough for understanding how Large Language Models (LLMs) leverage in-context learning for retrieval augmentation.
2. Deep Insight into Model Mechanisms: The research provides a detailed analysis of how "in-context" attention heads specialize in instruction comprehension and information retrieval. It also demonstrates how these heads influence the answer generation process, significantly enhancing our understanding of LLMs' internal workings.
3. Foundational for Interpretable RAG Systems: The paper lays the groundwork for more transparent and trustworthy Retrieval-Augmented Generation (RAG) systems by offering preliminary methods for tracing source information in retrieval-augmented LLMs. This is a vital step towards building safer and more controllable AI.

Weaknesses
1. Limited Scope of Investigation: The study primarily focuses on the role of attention heads. It does not extensively explore the synergistic interactions between attention heads and other model components (such as MLP modules or knowledge neurons) in inducing specific functions, leaving this for future research.
2. Untested in Cross-Modal or Multi-Task Scenarios: While the paper demonstrates the transferability of attention heads across different datasets, its analysis mainly concentrates on unimodal question-answering tasks. It does not investigate whether these mechanisms apply similarly in more complex cross-modal or multi-task learning environments.
3. Potential for Malicious Exploitation: The paper acknowledges a potential risk that the identified attention heads could be exploited to induce malicious behavior. While not a flaw in the research methodology itself, it highlights an ethical and safety challenge associated with such deep insights into model control.

---

> ### Author Rebuttal · Authors · 2025-07-30
>
> We sincerely thank the reviewer for the constructive and positive feedback and the time spent on our manuscript. We are encouraged that our approach is seen as a crucial breakthrough in understanding how Large Language Models leverage in-context learning specifically for retrieval augmentation. Furthermore, we are grateful that the foundational nature of our preliminary methods for tracing source information within retrieval-augmented LLMs is appreciated as a vital step toward developing more interpretable, transparent, and ultimately safer AI systems. In the following, we provide detailed responses to the reviewers’ concerns.
>
> **[Methodological specificity and generalizability of attribution method, along with its mathematical definitions and computational procedures]**: We thank the reviewer for raising this important question. We would like to clarify that our study is intentionally designed to be **agnostic to the specific choice of feature attribution technique** and, as such, is compatible with a broad class of attribution methods. As outlined in Section 3 (L82–94), we carefully considered numerous existing attribution techniques but found many to be limited by issues such as **scalability or faithfulness** to the model’s internal mechanisms. Since our goal involves tracing and filtering attention heads across large datasets, our chosen method has to be both computationally efficient and rigorously faithful in attributing the model’s behavior.
>
> Rather than developing a novel attribution method, we selected AttnLRP [6] (detailed in Section 3.1, L105–113) due to its well-founded mathematical basis and suitability for our setting. AttnLRP is a (modified) gradient-based method **requiring only a single backward pass** to assign relevance values to all heads, which offers significant computational advantages and takes the entire computational path into account. Its core principle involves approximating the contribution of neurons to the final output logit through Jacobian computations, while specifically correcting gradient distortions introduced by non-linearities common in large language models. For instance, layer normalization notoriously disrupts gradient flow [1], but AttnLRP’s formulation carefully linearizes this operation by treating normalized vectors as constants by detaching them from the computational graph, as rigorously proven in their original work. Relevance values can subsequently be obtained directly from the `.grad` attributes of the corresponding tensors in the PyTorch implementation. We acknowledge that readers may benefit from a concise introduction to AttnLRP. To address this, we are considering adding a brief overview section in the Appendix. For a more detailed treatment, we refer readers to the original AttnLRP paper [6].
>
> We further illustrate the need for efficiency by comparing with other methods such as path patching [2] that requires one clean and one corrupted forward pass per head (plus an additional backward pass).  With 1024 attention heads, this results in 2048 forward passes per sample. On a **small test dataset** of NQ-Swap, where a single run takes approximately 15 minutes on NVIDIA GeForce RTX 4090, this would amount to **0.25h × 2048 ≈ 31.3 days, making such methods infeasible at scale**.
>
> **[Universality of insights and limitations in application, including robustness and generalizability across different Transformer architectures, different languages than English, and other types of ICL tasks]**: Our study generalizes beyond language modeling and is **applicable to all Transformer-based architectures**, including vision models, and is agnostic to the attribution method as well as. To further highlight the generalizability of our approach, we conducted experiments on **Machine Translation on the OPUS Europarl dataset (on French, Spanish, English)**. In these experiments, we identified task-specific heads responsible for translation in a **zero-shot manner**. We then compacted these heads into a feature vector representation and patched them into different prompts (again in a zero-shot fashion) successfully inducing translation behavior into various languages. To evaluate the effectiveness of this approach, we compared it against two baselines: (1) directly prompting the model with “Translate the sentence into [LANGUAGE]:” and (2) randomly patching attention heads. Our method matched the BLEU score of the explicit prompting baseline (implemented with SacreBLEU, 33 points on Es-Fr, 29 points on Fr-En), demonstrating that the patched task heads reliably induce the desired translation behavior. Notably, this was achieved using only 15 of the 1024 attention heads in the LLama 3.1-8B-Instruct model, highlighting the preciseness of our approach. Random patching, on the other hand, results in a BLEU score of close to zero. **We believe these results substantially strengthen our paper and we thank the reviewer** for suggesting to include machine translation experiments.
>
> Furthermore, we would like to highlight that **QA remains one of the most prevalent and impactful applications of LLMs**, which motivated our decision to focus on this task. While prior studies on attention heads have largely been restricted to simple **toy settings** as we mentioned in our related work section, to the best of our knowledge, we are among the first to scale this line of analysis to real world datasets. **This represents a substantial advancement in the field.** While we are currently unable to provide an analysis on multimodal image-text data, we believe our results are likely to generalize to vision-language models (VLMs). This is because VLMs typically retain the core architecture of language models pretrained on text, with the addition of a vision encoder or projector that maps image inputs into soft text tokens. The underlying LLM remains largely unchanged [3]. We plan to add this as a limitation and an exciting direction for future research.
>
> **[Absoluteness of classifications of attention heads]**: Thank you for the insightful question. As shown in Figure 3, we plotted the in-context scores of all attention heads in sorted order. The results reveal a continuous spectrum ranging from in-context (positive score) to parametric behavior (negative score). Notably, many heads fall in the middle of this distribution, with average scores slightly above zero. This suggests that a substantial number of heads are not specialized exclusively for either in-context or parametric roles and their mixed contributions effectively cancel out in the scoring. We agree that further analysis of these intermediate heads is a compelling direction for future work, and we plan to elaborate on this point in our manuscript.
>
> **[Interaction with MLP modules]**: We agree that investigating the interplay between attention heads and MLP layers is a highly valuable direction. We hypothesize that parametric heads ultimately extract attributes from token positions where MLP layers have previously enriched the representations. Analyzing this interaction is considerably more complex due to the polysemantic nature of concepts encoded in MLPs, hence the growing body of work on sparse autoencoders (SAEs) [4].
>
> Due to the limited scope and space constraints of our current manuscript, we did not explore this aspect in depth. However, we have acknowledged this limitation already explicitly in the current version. It is worth noting that many recent studies have focused **exclusively on analyzing attention heads** [5] and likewise **omitted the study of MLPs**. In this context, we believe our in-depth analysis of attention heads already constitutes a substantial and meaningful contribution to the field.
>
> **[Potential for misuse/exploitation]**: As you rightly mentioned, this is a general limitation tied to deep interpretability, rather than a weakness of our method. We already acknowledged this as a potential risk in the main paper.
>
> We thank the reviewer again for the time spent engaging with our work and we hope our responses have clarified the points raised.
>
> References:
>
> [1] Xu, Jingjing, et al. "Understanding and improving layer normalization." Advances in neural information processing systems 32 (2019).
>
> [2] Goldowsky-Dill, Nicholas, et al. "Localizing model behavior with path patching." arXiv preprint arXiv:2304.05969 (2023).
>
> [3] Liu, Haotian, et al. "Visual instruction tuning." Advances in neural information processing systems 36 (2023)
>
> [4] Gao, Leo, et al. "Scaling and evaluating sparse autoencoders." arXiv preprint arXiv:2406.04093 (2024).
>
> [5] Wu, Wenhao, et al. "Retrieval Head Mechanistically Explains Long-Context Factuality." ICLR (2025).
>
> [6] Achtibat, Reduan, et al. "AttnLRP: Attention-Aware Layer-Wise Relevance Propagation for Transformers." International Conference on Machine Learning. PMLR, 2024.

---

> > ### Author Response · Authors · 2025-08-05
> >
> > Dear Reviewer n2tY,
> >
> > we have replied to your questions. As the deadline is approaching, could you please respond at your earliest?
> >
> > Looking forward to further discussion and also happy to address any additional questions you may have.

---

### Official Review · Reviewer_J8iq · 2025-07-05

**Clarity:** 3
**Significance:** 2
**Originality:** 2
**Rating:** 3
**Confidence:** 3

**Summary:**

This work presents probing results aimed at understanding the inner workings of large language models (LLMs) in the context of question answering. Building on the existing AttnLRP method, the authors identify three types of attention heads—in-context (input-dependent), parametric (internal knowledge-centric), and retrieval-based (copying)—and explore their respective roles in solving QA tasks using the NQ-Swap and TriviaQA datasets.

In Section 5, the authors argue that their method outperforms both a random baseline and Attention Weight Recall (AWR), thereby justifying the use of AttnLRP. In Section 6, they observe that removing in-context heads leads to performance degradation in closed-book settings. Based on this finding, they conjecture that in-context heads not only process contextual information but also play a role in interpreting the so-called “intensional frame”—the semantic structure imposed by the task instruction. Within this framework, the authors suggest that task heads—although it remains unclear whether these are a subset of in-context heads or merely share some overlapping functions—are responsible for encoding the intensional frame (e.g., the subject–relation–object structure), while parametric heads encode information about the subject entity.
This hypothesis is empirically supported by experiments involving function vectors derived from the corresponding attention heads. In Section 7, the authors further train a linear probe to examine whether the representations involved in the retrieval mechanism are linearly separable.

**Questions:**

[Follow‑up questions after rebuttal]

1. About the (dataset and task) coverage of this work

I acknowledge that this work aims to address more complex cases by considering two QA datasets. However, I do not believe this is sufficient to claim that it covers most real‑world scenarios. As the authors themselves note, the evaluation is limited to cases involving short‑form answers. The overall storyline appears to discuss the inner workings of ICL from a general perspective, without restricting the discussion solely to QA datasets. (Although, in the abstract, the authors mention that “we shed light on the mechanism of in‑context retrieval augmentation for question answering,” the introduction, in my reading, presents the discussion in the broader context of general in‑context learning rather than only QA.) If this interpretation is correct, it would be difficult for most readers to agree that evaluation on just two QA datasets is sufficient to support the claim that the findings will generalize to all possible uses of LLMs, including tasks requiring long‑form answer generation.

2. In‑Context Learning vs. In‑Context Retrieval Augmentation

Continuing from the above discussion, what is the exact difference between in‑context learning and in‑context retrieval augmentation?
As these terms are used interchangeably in the draft, it would be beneficial to clearly define their commonalities and distinctions. My understanding is that in‑context learning is the broader concept, whereas in‑context retrieval augmentation—the main focus of your work—represents a narrower, more specific instantiation of it.
In relation to your experimental settings and target tasks, a clearer and more consistent use of these terms would help contextualize the contribution and scope of your work.
I am still somewhat unclear on whether this work aims to generalize its findings from QA tasks to broader use cases, or whether it is intended to focus exclusively on QA tasks.
Please clarify this point.

3. Insufficient baselines (or [Unsuitable baselines])

Although the authors defended their choice of using only one baseline with the following justification: “Regarding baselines, we note that AWR [4] is, to the best of our knowledge, the most recent and relevant method capable of extending to multi‑token settings. Other closely related work [5] is limited to single‑token responses and does not provide code, which makes direct comparisons difficult.”

I appreciate their response, but I remain unconvinced by this justification.
As the authors themselves noted, the target datasets—TriviaQA and NQ‑Swap—require relatively short‑form answers.
For example, in the TriviaQA dataset (see example instances in https://huggingface.co/datasets/mandarjoshi/trivia_qa/), answers often consist of only two words.
Given this, I believe the authors could make a reasonable effort to adapt existing baselines to these settings without substantial modifications.
Alternatively, they could consider evaluating on subsets of the datasets containing instances with single‑word answers to allow a fairer comparison with existing work.
In summary, I believe the contribution of this work could and should be directly compared to prior approaches to more clearly highlight its unique contributions.

4. Discussion about [Unclear novel contributions wrt. prior studies], [Usage of special entity-based dataset], [Knowledge conflict results]

Thank you for the clarification.
It would help readers better understand the context if the corresponding parts were revised to guide them more clearly in line with your intentions.
In other words, the writing could be further refined to improve the clarity and effectiveness of idea communication.

5. Clear definitions of some key words (continued on the question 2)

Keywords used in this work, such as function vectors and in‑context heads, are already widely adopted in the related literature. However, in some cases, the definitions of these terms in this work appear to differ from their usage in prior studies. It would therefore be highly beneficial to clearly define the intended semantics of these keywords in the draft.
For example, are the function vectors mentioned in this work equivalent to those described in "Todd et al., Function Vectors in Large Language Models, ICLR 2024" [62]? If not, what are the precise differences?
This point is also connected to the discussion in the authors’ response under the section [Unclear relations between heads].

**Ethical Concerns:**

["NO or VERY MINOR ethics concerns only"]

**Final Justification:**

As the discussion phase is still ongoing, the final justification will be determined after its conclusion.
At this stage, I acknowledge the authors’ responses; however, I believe that the score I have assigned remains reasonable.

**Limitations:**

[Follow‑up discussion on the limitations of this work after rebuttal]

I believe that the limitation discussed in the draft—namely, that the proposed method is applicable only to attention heads and not to other components of Transformers—is not sufficient to meaningfully guide revisions of this work or inform related future research.
A more substantive limitation is the narrow coverage in terms of tasks and methodologies, which should be addressed.
For instance, investigating whether the findings can be generalized to more complex QA tasks or to entirely different task types would be both meaningful and desirable, potentially as part of the revision of this work itself.

In this context, I appreciate the authors’ efforts during the rebuttal process to report the [In‑context MT results]. I agree that the inclusion of such additional experiments can strengthen the research.

**Quality:**

2

**Strengths And Weaknesses:**

Strengths
- The paper presents a series of analyses grounded in existing and reliable techniques, including AttnLRP.
- It attempts to verify that specific attention heads in LLMs play distinct roles in processing question answering tasks.

Weaknesses
- It remains unclear what novel contributions this work offers. The study builds on existing methods and identifies attention head categories—such as in-context and parametric heads—that are already well known. What distinguishes this work from prior studies, beyond reinforcing existing findings?
- Some methodological details are ambiguous. For instance, why is a special entity-based dataset used specifically in Section 6 but not in other experiments? Additionally, the relationships between key concepts such as (1) in-context heads, (2) task heads, and (3) retrieval heads are not clearly defined. Several terms are introduced without precise explanation, which may confuse readers.
- Although Section 5 provides empirical evidence that the adopted method outperforms a random baseline and AWR, it is not evident that this is the most suitable technique for the type of analysis conducted in this work. Given the rich body of literature on probing LLMs, further justification or comparisons with additional baselines would strengthen the claim.
- The attention head types examined in this work (e.g., in-context and parametric heads) are already well-established in related literature. Consequently, the contribution of this study is unclear, especially since the analysis closely resembles previous work. Furthermore, the scope of the analysis is quite limited—it covers only two QA datasets and three relatively small LLMs—raising concerns about the generalizability of the findings and whether the models analyzed are representative of truly large-scale, capable LLMs.

---

> ### Author Rebuttal · Authors · 2025-07-30
>
> We sincerely thank the reviewer for the constructive feedback and for the time spent evaluating our manuscript. We are grateful for the recognition of our rigorous analysis uncovering the distinct roles that attention heads in LLMs fulfill. However, we respectfully offer the following clarifications to address the concerns raised, which we hope will highlight the novelty of our contribution.
>
> **[Unclear novel contributions wrt. prior studies]**: While prior studies have indeed identified heads with behavior similar to in-context and parametric heads, as we acknowledged and discussed in our related work section, our findings substantially advance the understanding of these mechanisms in several important ways:
> - Prior studies mostly focused on **narrow domains where the answer is composed of a single token, far from realistic use cases**. Here, we analyse popular open-domain QA datasets such as NQ-Swap & TriviaQA along with a human biography datasets that we curate, **which have short-form answers and better reflect the complexity of real-world language tasks**.
> - For the first time, we show that **in-context heads are considerably more complex than previously thought and can specialize to task or retrieval heads** which provides a finer-grained categorization than is available in existing literature. We also empirically demonstrate their causal roles in shaping the model’s internal representation (via intensional frames) along with parametric heads.
> - Prior works were restricted to **simple toy tasks due to the computational costs of the employed methods**. Our novel approach of heads filtering, along with the efficiency of AttnLRP allows **better scaling and enables extraction of steerable function vectors (FVs) in a zero-shot manner for complex tasks**, as described in Section 6.2. To substantiate how our method also generalizes beyond QA, we have included additional preliminary experiments on machine translation, detailed below.
> - We also introduce **a novel post-hoc procedure to trace both contextual and parametric knowledge usage** in LLMs, thus enabling the identification of potential hallucinations, knowledge conflicts and citations (Section 7). We have added additional experiments demonstrating that our method can effectively mitigate knowledge conflicts. While we present our findings as a preliminary step, we believe the results are promising and plan to develop this direction further in future work, especially since this line of work has become increasingly important in the community [1,2,3].
>
> **[Usage of special entity-based dataset]**: We would like to clarify that all major experiments (including the extraction of in-context, parametric, retrieval, and task heads)  are conducted on **realistic QA datasets**. The entity-based synthetic dataset is used solely for **controlled experiments** aimed at validating specific functional hypotheses: namely, that parametric heads encode entity attributes, task heads capture intensional frames, and retrieval heads copy answer tokens verbatim. Such fine-grained ground truth supervision is difficult to attain with realistic datasets, making controlled settings essential for isolating these behaviors, especially since entities might not possess the same attributes that we want to investigate across datapoints. We appreciate the reviewer’s feedback and will revise the manuscript to make this distinction clearer.
>
> **[Unclear relations between heads]**: We appreciate the reviewer’s request for clearer distinctions between in-context, task, and retrieval heads. These distinctions are visually and conceptually defined in Figure 1, in the introduction (L36-37), and further explained in Section 6.1. That being said, we agree that their relationship could be emphasized more strongly, and we will revise the manuscript to clearly state that **in-context heads can specialize into either task or retrieval heads, depending on whether they encode intensional frames or retrieve contextual facts**.
>
> **[Unsuitable baselines]**: Regarding baselines, we note that AWR [4] is to the best of our knowledge, the most recent and relevant method capable of extending to multi-token settings. Other closely related work [5] is limited to single-token responses and **does not provide code**, which makes direct comparisons difficult.
>
> While we outperform random and AWR baselines, our primary focus here is to enable a fine-grained analysis of attention heads in large-scale, realistic datasets, targeting heads that causally affect the answer generation process during in-context retrieval augmentation. We would like to emphasize, however, that **our method is agnostic to the choice of feature attribution technique and is compatible with a wide range of methods**. As detailed in Section 3 (L82–94), many existing approaches are not well-suited for our setting due to **limitations in scalability or faithfulness**. In particular, our need to trace and filter attention heads across large datasets necessitates a method that is both computationally efficient and faithful in attributing model behavior. For this reason, we employ AttnLRP, as justified in Section 3.1 (L105–113).
>
> To illustrate the computational challenge with less efficient methods, consider path patching [6], which roughly requires one clean and one corrupted forward pass per head. With 1024 attention heads, this results in 2048 forward passes per sample. On a **smaller test dataset** of NQ-Swap, where a single run takes approximately 15 minutes on NVIDIA GeForce RTX 4090, this would amount to **0.25h × 2048 ≈ 31.3 days, making such methods infeasible at scale**.
>
> **[Limited scope of analysis]**: **Given relevant related work, we can confidently state that the scope of our experiments goes beyond the usual extent of the current state of the art.** Compared to toy settings and single-token answers used in prior works, our analysis on two open-domain QA datasets along with the curated biography dataset represent a substantial step forward, especially wrt. a more realistic use case. We acknowledge that we don’t have access to very large-scale models due to computational constraints. However considering our experimental settings and fine-grained analysis, we would like to point out **our analysis of 7-9B parameter models is much bigger compared to those used in [5] relevant to this work** (the authors mostly experimented on toy data with single-token answers with short contexts), which is made feasible by our careful choice of an efficient and faithful attribution method. As it stands, we believe our contribution is also applicable to industrial-scale LLM analysis.
>
> **[In-context MT results]**: To further highlight the generalizability of our approach, we conducted experiments on machine translation on the OPUS Europarl dataset (on French, Spanish, English). In these experiments, we identified task-specific heads responsible for translation in a zero-shot manner. We then compacted these heads into a feature vector representation and patched them into different prompts (again in a zero-shot fashion) successfully inducing translation behavior into various languages. To evaluate the effectiveness of this approach, we compared it against two baselines: (1) directly prompting the model with “Translate the sentence into [LANGUAGE]:” and (2) randomly patching attention heads. Our method **matched the BLEU score of the explicit prompting baseline**  (implemented with SacreBLEU, 33 points on Es-Fr, 29 points on Fr-En), demonstrating that the patched task heads reliably induce the desired translation behavior. Notably, this was achieved using only 15 of the 1024 attention heads in the LLama 3.1-8B-Instruct model, **highlighting the preciseness of our approach**. Random patching has a BLEU score close to zero. **We believe these results substantially strengthen the contribution of our paper.**
>
> **[Knowledge conflict results]**: We computed the accuracy of LLama 3.1-8B-Instruct on the NQ-Swap with counterfactual contexts. Computing the recall score on the gold answer instead, the model is clearly distracted by the wrong contexts achieving only 9% accuracy. Removing the top 60 in-context heads reduces the reliance on the wrong context pushing the score to 35% which is similar to the closed-book accuracy of the model. Conversely, computing the recall score for counterfactual answers i.e. reliance on the counterfactual inputs, the model achieves 68% accuracy without ablation. Ablating the top 100 parametric heads reduces the reliance on internal knowledge and the model is more likely to accept the counterfactual context increasing the score to 76%.
>
> We sincerely thank the reviewer for the time spent engaging with our work and hope we could clarify all points raised.
>
> References:
>
> [1] Xu, Rongwu, et al. "Knowledge conflicts for LLMs: A survey." arXiv preprint arXiv:2403.08319 (2024).
>
> [2] Gao, Tianyu, et al. "Enabling large language models to generate text with citations." EMNLP (2023).
>
> [3] Qi, Jirui, et al. "Model internals-based answer attribution for trustworthy retrieval-augmented generation." EMNLP (2024).
>
> [4] Wu, Wenhao, et al. "Retrieval head mechanistically explains long-context factuality." ICLR (2025).
>
> [5] Jin, Zhuoran, et al. "Cutting off the head ends the conflict: A mechanism for interpreting and mitigating knowledge conflicts in language models." Findings of ACL (2024).
>
> [6] Goldowsky-Dill, Nicholas, et al. "Localizing model behavior with path patching." arXiv preprint arXiv:2304.05969 (2023).

---

> > ### Author Response · Authors · 2025-08-05
> >
> > Dear Reviewer J8iq,
> >
> > we have replied to your questions. As the deadline is approaching, could you please respond at your earliest?
> >
> > Looking forward to further discussion and also happy to address any additional questions you may have.

---

### Note · Authors · 2025-08-13

We sincerely thank the reviewers for their excellent comments and suggestions. We’re encouraged that they found our contribution on the mechanistic analysis of in-context retrieval augmentation on question answering task to be novel (n2tY) and backed up by a series of extensive and detailed experiments (J8iq, n2tY, FEvZ) that show strong empirical evidence (FevZ, ZnzS). We are particularly grateful that all reviewers acknowledge our attempt at understanding distinct roles of attention heads. They found it to be interesting (FEvZ) and provide valuable insights into the nature of in-context retrieval augmentation (n2tY, FEvZ).

Our findings not only further our understanding of in-context retrieval augmentation, but as reviewer n2tY noted, “lays the groundwork for more transparent and trustworthy RAG systems”. As such, our work has promising implications for a safer and more controllable AI, namely enabling causal tracing of source information (n2tY) and supporting the prevention of knowledge conflicts (ZnzS). Both of which play important roles in relation to hallucinations.

We appreciate reviewers J8iq & ZnzS’s recognition on our efforts to address more complex cases compared to prior studies. We are happy that reviewers FevZ and J8iq found our approach to be sound and principled (FevZ), while also being grounded in reliable techniques (J8iq). Furthermore, we appreciate that reviewer ZnzS found our work to be solid and provide comprehensive coverage, with effective visual presentation.

We address the reviewers’ questions and comments in the individual response boxes. To sum up, we will incorporate their feedback by:
- Adding the suggested clarifications to the text and improving the presentation of our paper (fix notation and figure issues).
- Incorporate the missing references into the related work section and expand to better contextualize the key differences of our work with prior studies.
- Enrich our results section with additional results on knowledge conflicts and in-context machine translation, along with additional analyses containing answers to the questions. Please note that the main aim of our work is to “shed light on the mechanism of in-context retrieval augmentation for question answering”, as we stated in the abstract.

We thank all the reviewers once more for their time and constructive feedback.

---

### Decision · Program_Chairs · 2025-09-17

**Decision:**

Accept (poster)

**Comment:**

The paper mechanistically analyzes retrieval augmentation -- how different heads perform different functions such as retrieving from the context vs. from the memory.

While the reviewers were on the borderline (two borderline rejects), the paper garnered one strong advocate (`ZnzS `). Based on my further impression of the paper, I recommend acceptance.

More details below:
- `J8iq` raised concerns about novelty -- that this paper combines many ideas, but lacks a significantly novel contribution. `ZnzS ` says that although each individual idea exists prior to this paper, the application is novel and brings out a new finding. I agree with this characterization.
- `J8iq` also brought up the fact that the study focused on two QA questions, and `FEvZ` says these are only one-token answer questions. `ZnzS` agrees with this concern especially in light of how the authors distinguish themselves from Jin et al., **I agree with the reviewers that this difference against Jin et al., cannot be claimed. Please edit your manuscript accordingly.**
- Both `ZnzS` and `J8iq ` brought up lack of contextualization against other works, including mechanistic interpretability literature, and a specific paper that is closely related but the authors dismissed it too casually (there was back and forth on this with `FEvz`, `ZnzS`). From my discussion, this did not seem to be debilitating in terms of the contributions of the paper. **Regardless this is very important: I strongly encourage the authors to do  do a thorough literature search, and rewrite how they contextualize these contributions against these other lines of related work. Please also cite and compare with Jin et al., prominently and accurately.**
- `FEvZ` has a main concern about the practicality of the result. I consider this as a subjective concern; there is possible future value in understanding the circuitry behind these behaviors. However, it may not hurt to dwell more on this and provide some suggestions at the end of the paper.